# Nucleotide-induced hyper-oligomerization inactivates transcription termination factor ρ

Bing Wang [1,5], Nelly Said [2,5], Tarek Hilal [2,3], Mark Finazzo [1], Markus C. Wahl [2,4] ✉ & Irina Artsimovitch [1] ✉

Bacterial RNA helicase ρ is a genome sentinel that terminates the synthesis of damaged and junk RNAs that are not translated by the ribosome. It is unclear how ρ is regulated during dormancy or stress, when translation is inefficient and RNAs are vulnerable to ρ-mediated release. We use cryogenic electron microscopy, biochemical, and genetic approaches to show that substitutions of residues in the connector between two ρ domains or ADP promote the formation of extended *Escherichia coli* ρ filaments. By contrast, (p)ppGpp induces the formation of transient ρ dodecamers. Our results demonstrate that ADP and (p)ppGpp nucleotides bound at subunit interfaces inhibit ρ ring closure that underpins the hexamer activation, thus favoring the assembly of inactive higher-order oligomers. Connector substitutions and antibiotics that inhibit RNA and protein syntheses trigger ρ aggregation in the cell. These and other recent data implicate aggregation as a widespread strategy to tune ρ activity.

All living cells possess mechanisms that silence the expression of potentially harmful or useless DNA. In bacteria, transcription factor Rho/ρ triggers the premature release of antisense, horizontally transferred, and untranslated RNAs[1–4]. ρ is a hexameric, ring-shaped helicase composed of two domains bridged by a flexible connector region[5]. The N-terminal domain (NTD) contains a primary RNA-binding site (PBS), the C-terminal domain (CTD) harbors a secondary RNA-binding site (SBS) and ATPase and helicase modules (Fig. 1a). An open-ring ρ binds to C-rich *rut* (*r*ho *ut*ilization) RNA sites via the PBS and then closes to trap RNA in the SBS, activating ρ ATPase and translocation along the RNA[6–8]. Alternatively, the open ring is recruited to the transcribing RNA polymerase (RNAP) through contacts to RNAP subunits and elongation factors NusA and NusG, then captures the nascent RNA via the PBS, and inactivates RNAP[9,10]. Both pathways have been shown to occur in vitro[11].

ρ surveils the nascent RNA to ensure that only those RNAs that are either translated by the ribosome or protected by dedicated anti-termination factors[1] are made. However, during slow growth or stress, when translation is inefficient, indiscriminate termination by ρ may be lethal; consistently, partial loss-of-function mutations in the *rho* gene enable *E. coli* survival on ethanol, which inhibits translation[12,13]. ρ cellular levels are held nearly constant by autoregulation[14], suggesting that ρ must be inhibited during stress. Recent studies revealed molecular mechanisms by which two stress-associated bacterial proteins, YihE and Rof, control ρ activity. These proteins directly bind the ρ NTD to block its interactions with RNA and/or RNAP, thereby inhibiting termination[15–17].

Here, we show that stress-associated nucleotides ADP and (p) ppGpp bind to the nucleotide-binding site in the CTD to trigger the formation of inactive ρ aggregates. We also show that changes in the

[1]Department of Microbiology and Center for RNA Biology, The Ohio State University, Columbus, OH, USA. [2]Freie Universität Berlin, Institute of Chemistry and Biochemistry, Laboratory of Structural Biochemistry, Takustr. 6, Berlin, Germany. [3]Freie Universität Berlin, Institute of Chemistry and Biochemistry, Research Center of Electron Microscopy and Core Facility BioSupraMol, Fabeckstr. 36a, Berlin, Germany. [4]Helmholtz-Zentrum Berlin für Materialien und Energie, Macromolecular Crystallography, Albert-Einstein-Str. 15, Berlin, Germany. [5]These authors contributed equally: Bing Wang, Nelly Said. ✉e-mail: markus.wahl@fu-berlin.de; artsimovitch.1@osu.edu

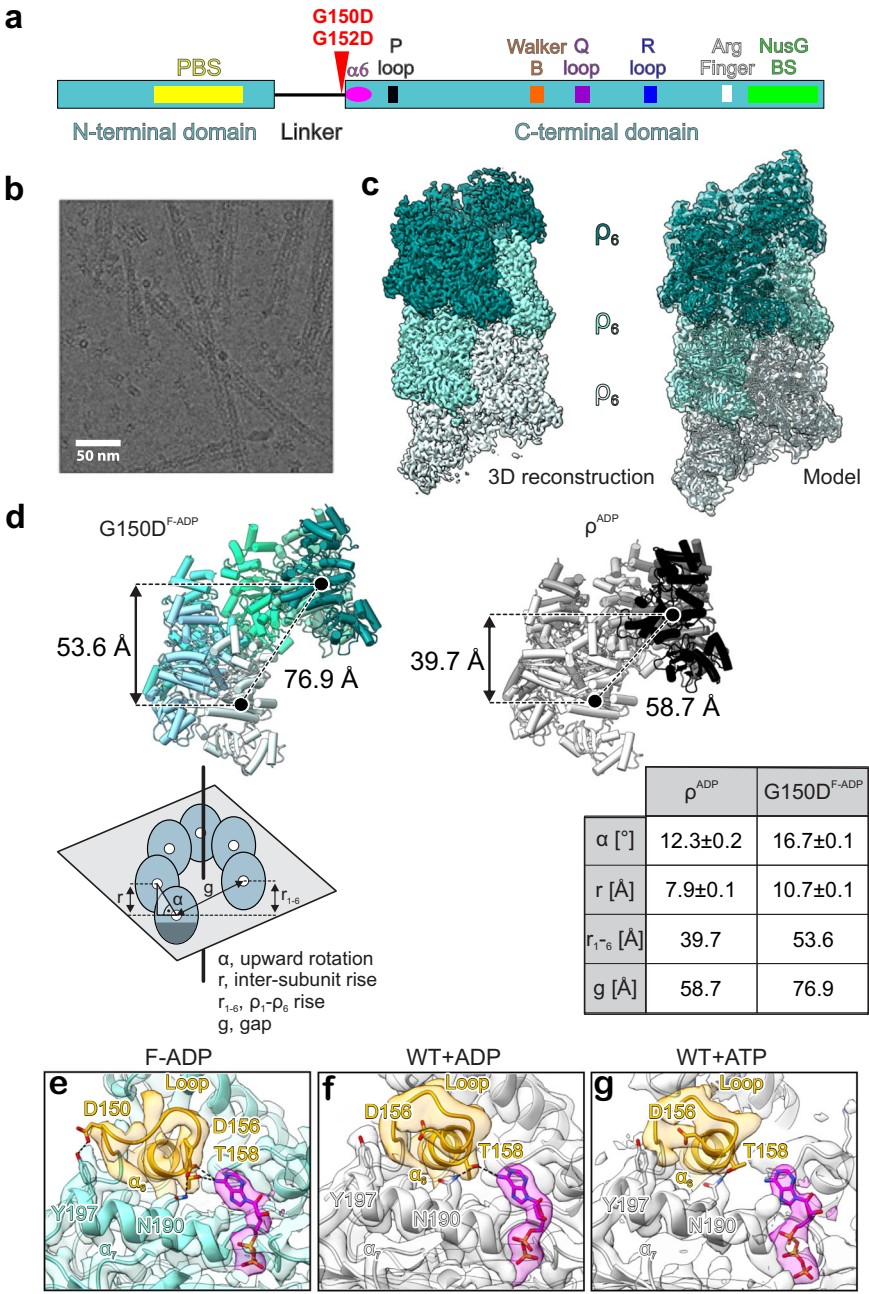

**Fig. 1 | *E. coli* ρ with substitutions in the connector region form extended filaments. a** Schematic diagram of ρ with key ATPase motifs, NusG-binding site, and positions of filament-inducing substitutions indicated. **b** Negatively stained in vitro filaments of ρ G150D•$H_8$. Scale bar, 50 nm. Representative micrograph from three independent data acquisitions. **c** CryoEM reconstruction of G150D•$H_8$ filaments. See Supplementary Figs. 1 and 2 for structural analysis of ρ G152D•$H_8$ filaments. Left, 3D reconstruction, a locally filtered map; Right, model of the G150D•$H_8$ filament in cartoon and semitransparent surface representation. **d** Comparison of WT and G150D hexamer geometries. The G150D$^{F-ADP}$ open ρ hexamer ring is extracted from the filament structure and compared to the untagged WT-ADP structure. Values of protomer-protomer distances and angles are derived from the draw_rotation axis script in Pymol. **e–g** Nucleotide-induced changes in the interaction network of the α5/α6 loop (residues 149-153 in orange) and helix α6 (residues 154–166 in orange) of G150D•$H_8$ filament bound to ADP, untagged WT bound to ADP or ATP. Corresponding 3D reconstructions are shown in semitransparent surface representation. **e** F-ADP, ADP-bound ρ G150D•$H_8$ filament (this study), (**f**) untagged WT hexamer bound to ADP (this study). **g** untagged WT hexamer bound to ATP (PDB ID: 6WA8). ADP and ATP nucleotides are shown in magenta. The partial unwinding of α6 repositions D156 close to the bound ADP in the F-ADP structure (**e**) enables the formation of additional interactions between D156 and T158 and D156 and ADP, respectively.

flexible linker that connects the two ρ domains promote nucleotide-independent ρ oligomerization.

## Results

### ρ connector mutants form filaments

Studies of poorly translated xenogeneic operons may reveal mechanisms of cellular adaptation to translational stress. The cell wall biosynthesis *waa* operon is silenced by ρ unless an antitermination factor RfaH is present[18]. Using genetic selection for Δ*rfaH* suppressors, we identified two changes in the ρ connector encoding G150D and G152D variants defective in termination (Fig. 1a). We hypothesized that G-to-D substitutions may rigidify the connector, disrupting communications between ρ domains and inhibiting ring closure[18]. We thus determined structures of G150D and G152D ρ proteins carrying a

C-terminal octahistidine-tag ($H_8$) using cryogenic electron microscopy (cryoEM). Strikingly, we found that both variants formed extended helical filaments (Fig. 1b and Supplementary Figs. 1 and 2) that trap ρ in an inactive state unable to interact with either RNA or RNAP; in the main text, we focus on the G150D ρ protein.

Analysis of the cryoEM reconstruction of a G150D filament revealed that 18 ρ protomers are well-defined in density even though 2D micrographs show that the filament extends far beyond (Fig. 1b, c). We prepared G150D in the presence of the ATP transition-state analog ADP•BeF$_3$ but while the density for ADP is visible in the ATP-binding pocket at each interface, the BeF$_3$ moiety is absent. The filament forms along a left-handed helical axis, similar to the wild-type (WT) ρ hexamer in an open-ring conformation reported by Skordalakes and Berger[5], who proposed that an increase in the pitch of the ρ ring would foster oligomerization. Consistently, in the G150D structure, the helical pitch increases, showing an 11 Å inter-subunit rise and an upward rotation of 17°, compared to 8 Å and 12° in WT ρ (Fig. 1d). This leads to an enlarged opening of the ring, thus allowing an additional hexamer to join.

By progressing into a filamentous structure, ρ G150D forms additional contacts between its NTD and CTD. The filament is strengthened by interactions of the very C-terminal α-helix (α16; residues 408–418 of one subunit (ρ$_1$) with the NTDs of protomers ρ$_7$ and ρ$_8$ located seven and eight subunits upstream the helical axis; Supplementary Fig. 3a, b). The last three α helices (α14-α16) of the ρ$_1$ CTD are inserted between the central (loop β1/β2-β2-β3-α4, residues 59–88) and C-terminal (loop β4/β5, residues 103–110) portions of the ρ$_7$ and ρ$_8$ NTDs. R87 of ρ$_7$ extends toward the loop between α15 and α16 of ρ$_1$ and approaches M405 and M415. Likewise, E106 of ρ$_8$ approaches K379 and K417 within α14 and α16 of ρ$_1$, respectively (Supplementary Fig. 3a, b).

Residue G150 is located within α5/α6 loop (residues 149–153) sandwiched between helix α7 (residues 184–198 that connect to the P-loop of the ATP pocket) and the helix-loop-helix region formed between α15 and α16. The α5/α6 loop is followed by helix α6 (residues 154–166), which not only faces the nucleobase of the bound nucleotide but also interacts with residues in α14-α16 (Fig. 1e and Supplementary Fig. 3a). Structural comparison to WT ρ bound to ADP and ATP (Fig. 1f, g and Supplementary Figs. 4 and 5) revealed that the G150D substitution induces major rearrangements of the intra-molecular network within α5/α6 loop and proximal regions without affecting the overall structure of the protomer. In G150D, a partial unwinding of α6 at its N-terminal end repositions several residues to form new interactions. D156, which points away from the nucleotide in WT ρ, now directly forms a hydrogen bond with the nucleobase of the bound ADP and contacts N190 within α7. In addition, D150 forms a hydrogen bond with Y197 (Fig. 1e–g). Thus, the G150D/G152D substitutions seem to restrict an otherwise very flexible arrangement of the nucleotide-binding site by tethering the α5/α6 loop more tightly to neighboring regions of the CTD and to the bound nucleotide.

## Wild-type ρ forms oligomers in vitro and in vivo

To determine whether purified WT and G150D ρ proteins form filaments in solution, we used pelleting assays, in which protein filaments sediment at 100,000 + g while monomers and smaller oligomers remain in the supernatant. Since filaments were observed on grids with $H_8$-tagged ρ variants, we first tested the sedimentation of untagged WT and G150D ρ proteins. As reported for ParM filaments[19], about half of the G150D ρ was in the pellet, whereas WT ρ remained soluble (Fig. 2a). Nucleotides are known to affect the formation of filaments[19,20], and the loss of BeF$_3$ suggests that G150D filaments preferentially bind ADP (Fig. 1e). We found that ADP strongly promoted aggregation of the WT ρ but had a lesser effect on the G150D ρ (Fig. 2a). Similar results were obtained with $H_8$-tagged proteins (Supplementary Fig. 6a, b). ADP-dependent aggregation may be common among ρ proteins:

*Pseudomonas aeruginosa* ρ was also pelleted in the presence of ADP (Supplementary Fig. 7).

Pelleting assays cannot distinguish between filaments and other large aggregates. To test whether ρ forms filaments as revealed by the structures, we designed "sensor" Cys residues that would be closely spaced in filaments and used bismaleimidoethane (BMOE) (Fig. 2b), a high-efficiency short-length (8.0 Å) cell-permeable sulfhydryl-to-sulfhydryl crosslinker, to assess filament formation[21]. We found that crosslinks readily formed ex vivo when BMOE was added to intact cells expressing $H_8$-tagged WT or G150D ρ with C84/C405 (X1; Supplementary Fig. 6c) or C106/C375 (X2; Supplementary Fig. 6d) substitutions from plasmids. The crosslinks were observed only in the presence of BMOE and formed a pattern consistent with a mixture of dodecamers and higher-order oligomers. Crosslinking of purified untagged ρ variants showed that mimicking the pelleting results (Fig. 2a), G150D$^{X1}$ formed oligomers in the absence of nucleotides, whereas WT$^{X1}$ was crosslinked only in the presence of ADP (Fig. 2c, d and Supplementary Fig. 6e). The S84C ρ did not form oligomers in the presence of BMOE (Fig. 2d and Supplementary Fig. 6c), demonstrating that two Cys residues were necessary for crosslinking. Conversely, pelleting assays indicated that the tendency to form filaments was reduced in a ρ variant in which residues E106 and Q378 that establish contacts at the filament interface are replaced by alanines (Supplementary Fig. 8).

In the gel, we observed ρ monomers (~ 55 kDa), dimers, and higher-order oligomers (Fig. 2c). To simplify the description, we assign ρ monomers to hexamers (H) and dimers to dodecamers (D), although they can represent smaller assemblies, *e.g.*, pentamers and 11-mers, respectively; we term all larger species filaments (F). A ladder of crosslinked monomers, up to at least nine species corresponding to 8+ complete ρ rings, is visible on the gel, whereas larger species (500 + kDa) could not be resolved. In each case, most easily seen with the dimers, we observed two species with different mobilities (Fig. 2c), which may have arisen from differences in LDS binding or which could reflect ρ oligomerization in two different constellations (see below).

Our failure to detect filaments on grids with untagged proteins suggested that the tag, which is adjacent to α16 making key filament-stabilizing contacts (see above), may favor filament formation. Indeed, a density attributable to the tag was present near the filament interface (Supplementary Fig. 6a); BMOE-mediated crosslinking showed that G150D$^{X1}$•$H_8$, but not G150D$^{X1}$, formed filaments at 50 nM (Supplementary Fig. 6f); and pelleting assays detected filament formation at lower concentrations in the presence of the $H_8$ tag (compare Fig. 2a and Supplementary Fig. 6b). We conclude that the $H_8$ tag stabilizes the filaments and promotes their formation at lower ρ concentrations. Critically, patterns of BMOE-crosslinked products were nearly identical for the WT and G150D ρ$^{X1}$ variants in the presence of ADP (Fig. 2c). Collectively, our results demonstrate that *E. coli* ρ forms filaments in the presence of ADP (Fig. 2a, c, d). Unlike WT ρ, G150D ρ forms filaments even in the absence of ADP, but the fraction of G150D filaments increases three-fold, from 18 to 56 %, upon the addition of ADP (Fig. 2d).

## ADP stabilizes an open ρ ring

The ρ ring dynamics is thought to be controlled by a molecular switch involving residues at the protomer interface[22]. An inhibitory cation-π interaction between R353 and W381 has been suggested to stabilize the open conformation and prevent productive contact between R366 (the Arg finger) and ATP[22]. The R353-W381 cation-π interaction in the F(filament)-ADP-bound state (Fig. 3a) is identical between neighboring protomers. In contrast, in the ATP-bound state, the R353-W381 contact seems to be more dynamic. While the density for W381 is well-defined at each of the protomer interfaces, the density for R353 is either not resolved or discontinuous, indicating higher flexibility or multiple conformations of R353. The cation-π interaction must be disrupted

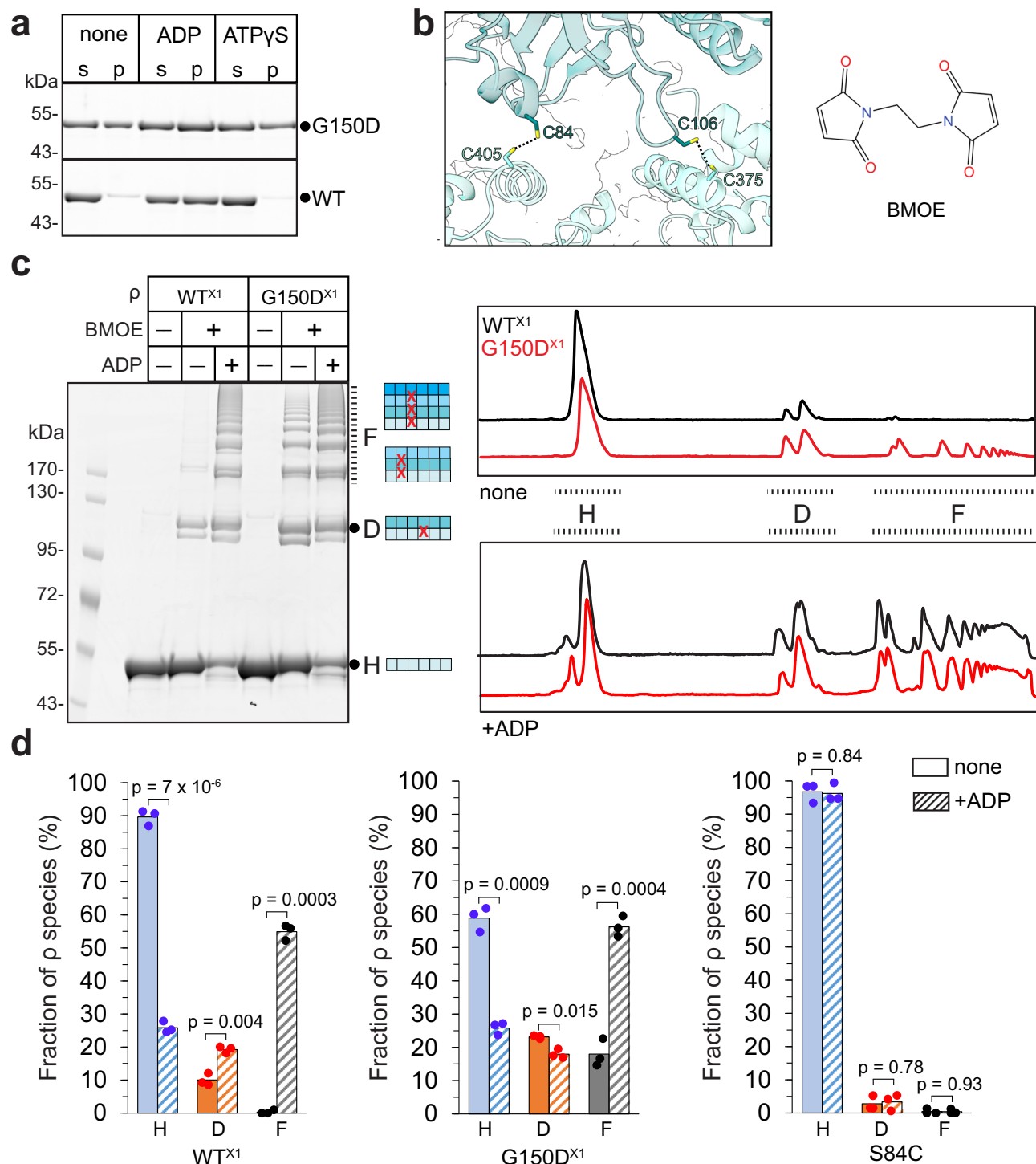

**Fig. 2 | Filament detection by pelleting and crosslinking. a** Nucleotide-dependent polymerization of ρ. Indicated untagged ρ variants (8 μM) were incubated with nucleotides (2 mM) for 10 min at room temperature and centrifuged at $160{,}000 \times g$ for 30 min at 20 °C. Supernatants (s) and solubilized pellets (p) were analyzed by LDS–PAGE and Coomassie Blue staining. Experiments were performed three times independently with similar results. Source data are provided as a Source Data file. **b** Sensor Cys pairs, C84/C405 (X1; 4.3 Å) and C106/C375 (X2; 3.3 Å), at the filament interface. **c** BMOE-mediated crosslinking of purified 1 μM untagged WT[X1] and G150D[X1] ρ in the absence or presence of 2 mM ADP. Left, reaction products were detected by LDS-PAGE and Coomassie Blue staining. Cartoons depict linearized single and stacked ρ rings that, when crosslinked (red X) via the engineered Cys residues at the

interface, can give rise to the observed products. Each ρ subunit can only be cross-linked to two neighbors, one from the ring above, and one from the ring below. Thus, a species that migrates as a dimer corresponds to at least two stacked ρ rings. H, D, and F indicate hexamers, dodecamers, and filaments, respectively. Right, lane profiles of the BMOE crosslinked species from the gel shown on the left for the WT (black lines) and G150D (red lines) ρ in the absence (top) and in the presence (bottom) of ADP. Experiments were performed three times independently with similar results. Source data are provided as a Source Data file. **d** Quantification of untagged ρ species after BMOE-mediated crosslinking in the absence or presence of ADP; two-tailed $T$ test assuming unequal variance was used to calculate $p$-values. Source data are provided as a Source Data file.

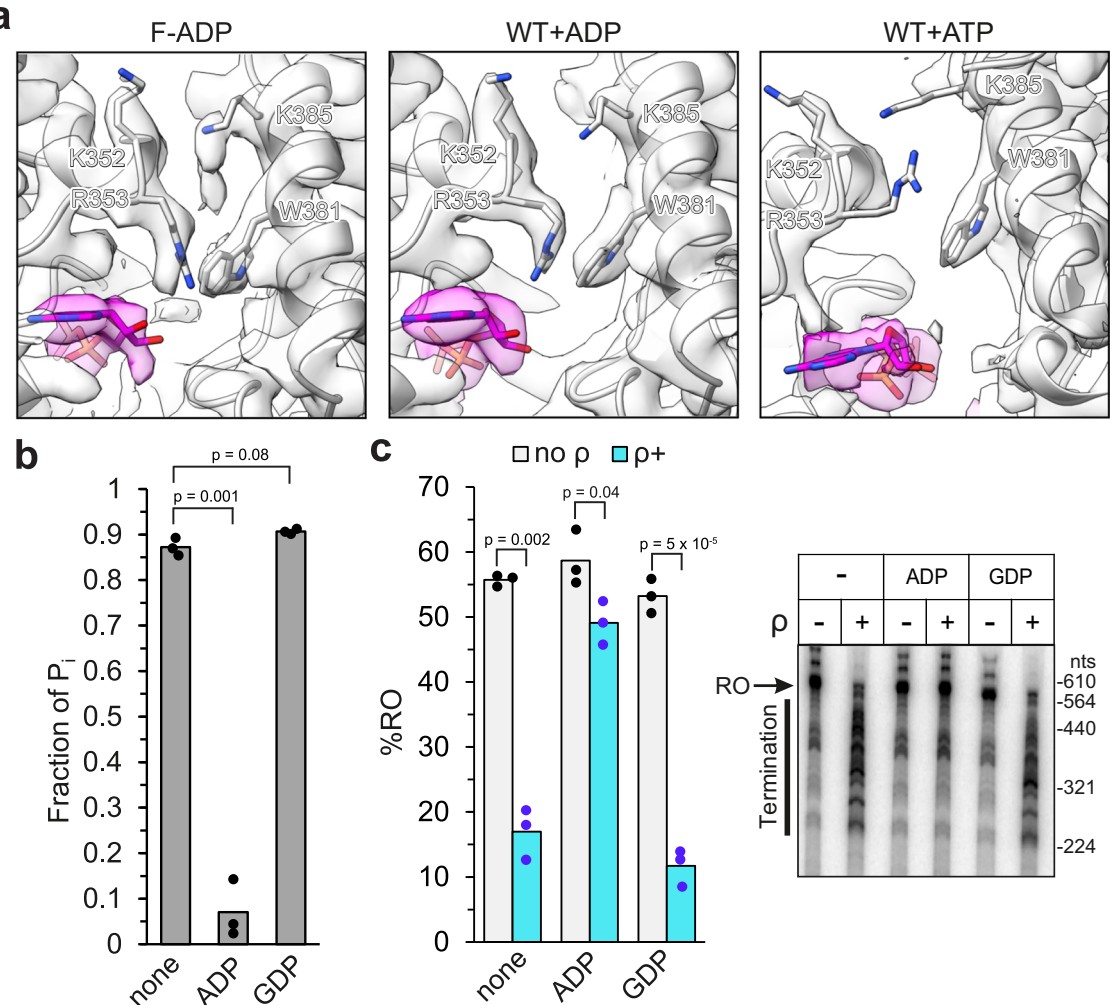

**Fig. 3 | ADP-stabilized filamentation compromises the activity of ρ. a** The ADP bound ρ G150D·H$_8$ filament (F-ADP) and untagged WT-ADP structures show a strong cation-π interaction between R353 and W381, whereas the interaction seems to be more flexible in the untagged WT-ATP structure (PDB ID: 6WA8). Here, the density for R353 is rather weak at several protomer interfaces, indicating that the R353 side chain conformation is not stabilized by W381. Experimental density is in semitransparent surface representation. **b** Effects of filamentation on ATPase activity. 1.5 µM untagged ρ was incubated with 2.5 mM nucleotides (GDP/ADP) or water (none) for 10 min at room temperature to facilitate filament formation. Then *rut* site-containing RNA (see "Methods") and ATP were added to 1.5 µM and 2 mM (containing 0.2 µCi/µL γ$^{32}$P-ATP) final concentrations, respectively. The reaction proceeded for 5 min before quenching by EDTA and products were resolved by

TLC. The fraction of P$_i$ was used as an indicator of ATPase activity. **c** Effects of filamentation on ρ-dependent termination. Nucleotide-stabilized filaments were formed by incubating 2.5 µM untagged ρ with 5 mM nucleotides (GDP/ADP) or water (none) for 10 min at room temperature. Then, the ρ-nucleotide mixture was diluted 5 times into halted radiolabeled transcription elongation complexes formed on the λ tR1 template (see "Methods"), and 150 µM NTPs were added to restart elongation. Positions of run-off (RO) transcript and ρ-dependent termination region are indicated on a representative urea-acrylamide gel. nts, nucleotides. The fraction of RO RNA was used to evaluate the termination activity of ρ. In vitro experiments were performed three times independently with similar results. Two-tailed *T* test assuming unequal variance was used to calculate *p*-values. Source data are provided as a Source Data file.

during ring closure[22], and the higher flexibility of R353 and W381 likely facilitates the underlying conformational changes.

We found that ADP inhibited ρ ATPase activity when present together with ATP (Fig. 3b), whereas GDP did not. Since ρ hydrolyzes both GTP and ATP[23], product inhibition is possible for ADP and GDP. However, as GDP does not affect the ρ ATPase activity, and as ATP is the preferred binding substrate over ADP in the presence of RNA[24], the observed inhibition likely reflects nucleotide-induced changes in ρ rather than product inhibition. We also found that ρ preincubation with ADP (under conditions that promote filament formation) compromised termination at a canonical λ tR1 signal, whereas preincubation with GDP had no deleterious effect (Fig. 3c).

**G150D substitution promotes hyper-oligomerization in vivo**

Unlike WT ρ, the G150D variant forms filaments in the absence of ADP in vitro (Fig. 2a, c). To determine whether G150D ρ is prone to

aggregation in the cell, we fractionated total cell extracts from strains carrying the WT or G150D *rho* chromosomal alleles on sucrose gradients; purified proteins were used as controls (Fig. 4a). We found that WT ρ was present predominantly as a hexamer in the cell, whereas G150D formed higher-order oligomers that were distributed along the gradient, with a substantial fraction (presumably long filaments) present in the pellet, confirming that G150D ρ polymerizes in the cell.

These extracts were prepared from (mildly stressed) stationary phase cells, which are expected to have elevated levels of ADP and (p) ppGpp, a stress alarmone that binds to many *E. coli* proteins including ρ[25]. We wondered if the G150D strain was more sensitive to acute stress induced by the accumulation of (p)ppGpp. We found that mupirocin (MUP), an inhibitor of isoleucyl-tRNA synthetase that triggers the stringent response[26], strongly inhibited the *rho* G150D strain but had lesser effects on the WT strain and a mutant with an IS2 insertion in the *rhoL* leader that leads to reduced levels of ρ[18] (Fig. 4b). Conversely, the

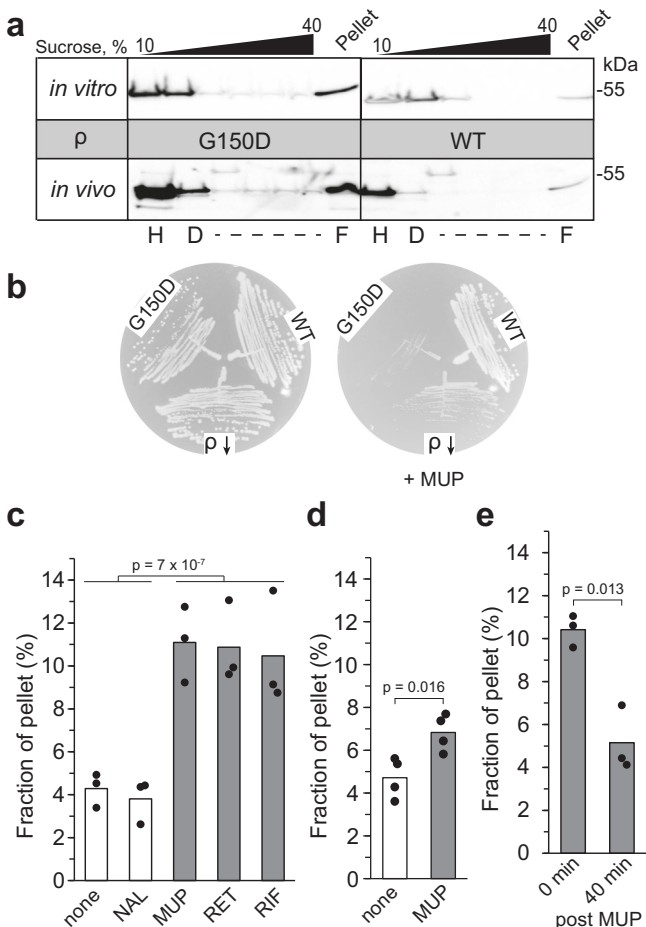

**Fig. 4 | ρ oligomers form and disperse in response to cellular cues. a** Sucrose density gradient analysis of WT and G150D ρ proteins in vitro (top) and in cells (bottom). ρ was detected by Western blotting with polyclonal anti-ρ antibodies. Tentative H, D, and F positions were assigned based on the sedimentation of known proteins. Experiments were performed three times independently with similar results. Source data are provided as a Source Data file. **b** WT, G150D, and ΩIS2 *rho* (ρ↓) cells were streaked on LB plates with or without MUP. Experiments were performed three times independently with similar results. **c–e** Quantification of in vivo pelleting assays (Supplementary Fig. 10a). The fraction of ρ pelleted at 110,000 x *g* (representing oligomers) was determined relative to the total amount of ρ in the sample (pellet + supernatant) using Western blotting. **c** ρ forms oligomers following a 30 min exposure to antibiotics that inhibit protein or RNA synthesis. —, none; NAL, nalidixic acid; MUP, mupirocin; RET, retapamulin; RIF, rifampicin. The *p*-value was calculated between (—, NAL) and (MUP, RET, RIF). **d** MUP effect in the ppGpp⁰ strain. **e** To investigate whether ρ oligomerization was reversible, cells were washed with LB following MUP treatment (carried out as in panel **c**) and incubated for 40 min in the absence of the antibiotic; control experiments demonstrated that cell growth did not resume after the 40 min recovery period (Supplementary Fig. 10c). All ρ variants used here are untagged. Two-tailed *T* test assuming unequal variance was used to determine *p*-values. Source data are provided as a Source Data file.

deletion of *relA* promoted the growth of G150D, but not of the WT strain (Supplementary Fig. 9a). The growth of the G150D strain was inhibited by a plasmid-borne (p)ppGpp synthetase RelA, but not its catalytically inactive D275G variant, whereas the WT strain and strains carrying other defective *rho* alleles were relatively resistant to both (Supplementary Fig. 9b). Taken together, these results show that the growth defects of the aggregation-prone G150D ρ are augmented by (p)ppGpp, perhaps through direct interaction with the alarmone; in vivo crosslinking identified ρ as one of many (p)ppGpp targets in *E. coli*[25].

Curiously, although G150D compromises *E. coli* ρ activity[18], some proteobacterial proteins have Asp at this position (Supplementary Fig. 3c). Our phylogenetic analysis revealed that residues at positions 150 and 152 are conserved only in γ-Proteobacteria and fall into two classes. While most ρ proteins have Gly at both positions, 28% have Asp/Glu at 150 and Arg/Lys at 152, a covariation that is likely to avoid the disruptive effects of the sole negative charge observed for the G150D or G152D variants studied here. We speculate that the R/K side chain at 152 restrains the Asp150 to keep the interaction network of the α5/α6 loop intact, resulting in a correct alignment of residues forming one part of the nucleotide-binding pocket.

**Wild-type ρ may hyper-oligomerize under stress**
We conjectured that ρ may form inactive oligomers during translational stress, when RNA becomes unprotected, or when RNA synthesis is arrested, triggering ρ release from RNAP. To test this hypothesis, we analyzed the ρ oligomeric state upon exposure of exponentially growing *E. coli* cells to antibiotics (Fig. 4c and Supplementary Fig. 10a, b). To stop protein synthesis, we used retapamulin (RET), which arrests the initiating ribosome[27], and MUP, which induces ribosome stalling during elongation[28]. To inhibit RNA synthesis, we used rifampicin (RIF), which blocks promoter escape[29]. As a control, we used nalidixic acid (NAL), an inhibitor of DNA gyrase[30]. We found that ρ was enriched in the pellet fraction following a 30-min exposure to MUP, RET, and RIF, but not after NAL or mock treatment (Fig. 4c). This effect was strongly reduced in the ppGpp⁰ strain and is thus at least in part attributable to ppGpp (Fig. 4d). We also observed that ρ partitions into the pellet under $Mg^{2+}$ starvation (Supplementary Fig. 10d), which promotes (p)ppGpp accumulation[31].

Some stress-induced oligomers, such as hibernating ribosomes or metabolic enzymes, are known to disassemble upon return to active growth[32], prompting us to ask whether ρ oligomerization is also reversible. We found that ρ aggregates triggered by MUP can be dispersed following antibiotic removal (Fig. 4e and Supplementary Fig. 10c). Collectively, our results show that *E. coli* ρ can reversibly hyper-oligomerize in response to cellular cues and suggest that temporary inactivation of ρ may be beneficial under stress.

**(p)ppGpp binds to the ATPase site and induces ρ oligomerization**
Consistent with our in vivo results, we found that ppGpp and pppGpp promoted ρ hyper-oligomerization in vitro, whereas GDP did not (Fig. 5a). However, the pattern was quite distinct from that observed with ADP, when long ladders of stacked hexamers were detected by crosslinking, comprising half of ρ (Figs. 2c and 5a). By contrast, with ppGpp, almost a third of crosslinked ρ was present in the dodecamer fraction and only ~2% formed larger oligomers (Fig. 5b). An increase in the dodecamer fraction was also observed with G150D ρ (Supplementary Fig. 11a), supporting a model in which *rhoG150D* hypersensitivity to (p)ppGpp in vivo (Fig. 4b and Supplementary Fig. 9b) may be due to extensive ρ aggregation.

To investigate the effects of (p)ppGpp on ρ, we solved a cryoEM structure of WT ρ bound to pppGpp (Supplementary Figs. 4 and 5); ppGpp did not stably bind to ρ under conditions of grid preparation. Structural analysis using single-particle cryoEM revealed one reconstruction of an open ρ hexamer with pppGpp bound at four ATP-binding pockets. Overall, the structure resembles other open-ring ρ structures and shows a similar conformation in the intra-molecular network within the loop region and helices α6 and α7 as compared to WT-ATP (Figs. 5c and 1g).

Our crosslinking shows that ADP, but not (p)ppGpp, promotes the formation of extended ρ filaments (Fig. 5a) and our structures suggest that ADP-filaments may be more stable than ppGpp-induced dodecamers. To evaluate this possibility, we pre-incubated WT ρ with ADP or ppGpp, followed by the addition of ATPγS for 30 min (Fig. 5d). We

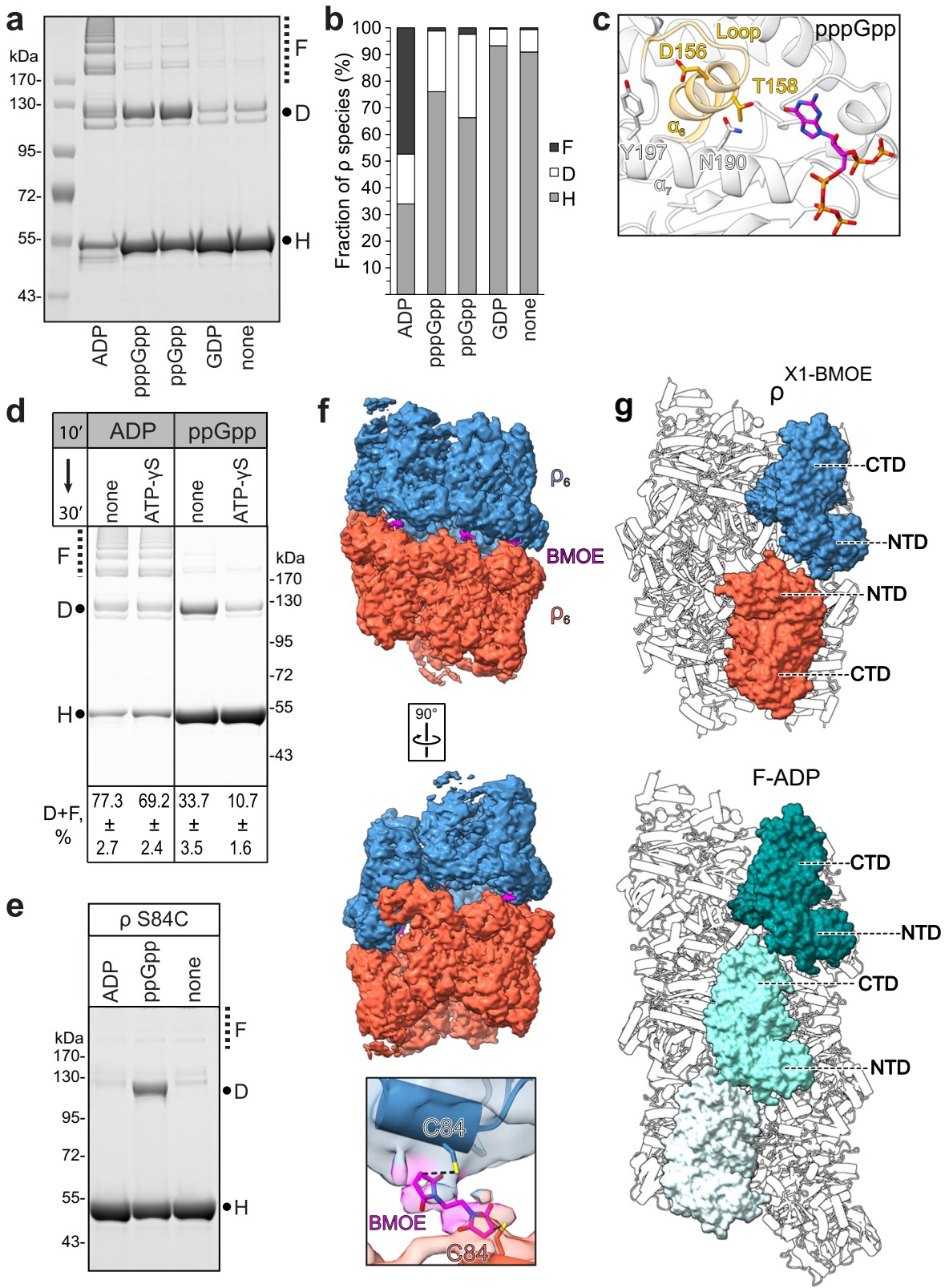

observed that ADP-stabilized filaments were only partially perturbed, whereas two-thirds of ppGpp-stabilized oligomers disappeared when challenged with ATPγS, explaining why ppGpp has only a modest negative effect on ATPase activity (Supplementary Fig. 11b). The effects of ppGpp on termination are complex because, unlike ADP and GDP, ppGpp directly binds to RNAP[33] and stimulates pausing, thereby increasing both intrinsic and ρ-dependent termination. The average nucleotide addition

rate by *E. coli* RNAP is decreased by 30% at 30 µM ppGpp, and ρ termination is increased at 25 µM ppGpp[34] whereas here we used 1 mM ppGpp. Our results show that the ppGpp effect on RNAP apparently dominates in the in vitro transcription assays, reducing readthrough at the λ tR1 signal with or without ρ (Supplementary Fig. 11c).

These results raise a possibility that ppGpp-stabilized dodecamers may have a distinct structure and are not precursors to filaments.

**Fig. 5 | (p)ppGpp promotes reversible ρ hyper-oligomerization. a** Detection of ρ oligomers by BMOE crosslinking in the presence of ADP, GDP, and (p)ppGpp; protein marker sizes are indicated on the left. Experiments were performed three times independently with similar results. Source data are provided as a Source Data file. **b** Distribution of oligomeric states induced by different nucleotides; reported as means, *n* = 3; see Source data for quantification. **c** Structure of WT ρ bound to pppGpp (this study). ρ binds pppGpp at its nucleotide-binding pocket. The α5/α6 loop (residues 149–153 in orange) and helix α6 (residues 154–166 in orange) are indicated. Side chain residues important for nucleotide coordination are shown as sticks; compared to Fig. 1e–g. **d** 2 mM ADP- and ppGpp-stabilized ρ polymers were formed in vitro for 10 min and challenged with a non-hydrolyzable ATP analog

(20 mM) for 30 min, followed by BMOE crosslinking. Percentages of ρ found in the D (dodecamer) and F (filament) fractions are shown as means ± SD; *n* = 4. Source data are provided as a Source Data file. **e** S84C ρ forms dodecamers in response to ppGpp. Experiments were performed three times independently with similar results; source data are provided as a Source Data file. **f** CryoEM map and structure of BMOE-crosslinked ρ[X1] variant in the presence of ppGpp; BMOE bridging two C84 residues is shown at the bottom. **g** Comparison of crosslinked ρ dodecamer to G150D[F-ADP]. Two ρ hexamers in ρ[X1-BMOE] structure are connected N-to-N. In contrast, in the G150D[F-ADP] structure, ρ subunits are continuously joined C-to-N. All ρ variants used here are untagged.

Indeed, we observed that ppGpp-induced crosslinking of S84C ρ (Fig. 5e) is incompatible with the filament structure. To corroborate this idea, we obtained a structure of ρ[X1] that was BMOE-crosslinked in the presence of ppGpp (Supplementary Figs. 12 and 13). In these stabilized complexes, we observed two open ρ rings stacked with their NTDs facing each other (N-to-N) and Cys84 residues bridged by BMOE (Fig. 5f, g). Our results demonstrate that, in the presence of (p)ppGpp, the majority of ρ forms transient "non-extendable" dodecamers that cannot give rise to filaments and co-exist with a minor fraction of the filament-competent state. The alternative C-to-N filament and N-to-N constellations might explain the two bands representing dodecamers observed after BMOE crosslinking (e.g., Fig. 2c).

Inefficient crosslinking of a ppGpp analog in vivo[25] suggested that ppGpp binds to ρ weakly; consistently, we could not accurately measure ppGpp affinity using ITC or DRaCALA. While the low binding affinity could safeguard ρ from fortuitous inactivation except during acute stress, how can ppGpp inhibit ρ in the presence of millimolar levels of ATP? To test this, we used purine nucleotide mixtures that correspond to "optimal" growth vs "stringent" conditions induced by MUP[26]. Our results show that ppGpp induces ρ oligomerization even in the presence of millimolar concentrations of ATP (Supplementary Fig. 11d). We also found that RNA, which promotes ρ ring closure, did not prevent ppGpp- or ADP-induced ρ oligomerization (Supplementary Fig. 14a).

The apparent distribution of ρ oligomeric states reflects their presence in solution and efficiency of crosslinking by BMOE and may be altered by Cys residues. To visualize oligomers of WT "native" ρ, we used sucrose gradient centrifugation. We observed that, in the absence of nucleotides, WT ρ was present in two fractions, consistent with a mixture of hexamers and dodecamers (Supplementary Fig. 14b), as reported previously[35,36]. In agreement with our crosslinking assays, ρ distribution was shifted toward longer oligomers in the presence of ppGpp and ADP (Supplementary Fig. 14b, c), with ADP favoring larger oligomers. We note that, in contrast to ADP-stabilized ρ filaments (Fig. 2a), ppGpp-stabilized dodecamers do not pellet in vitro (Supplementary Fig. 14d), suggesting that ρ aggregation in stressed cells (Fig. 4c) is promoted by yet-unknown cellular factors.

### ADP and (p)ppGpp promote hyper-oligomerization by changing ρ protomer interfaces

We carried out a detailed comparison of the binding modes of nucleotides[37] to WT hexamers and G150D filaments (Fig. 6a and Supplementary Fig. 11e). In the pppGpp-bound structure, we observed additional contacts of the 3′-pyrophosphate, with the δ-phosphate forming a hydrogen bond to the amino group of K367 and van der Waals interactions to E368 and E369. The nucleotide is further stabilized by G183 and K184 of one protomer and by R366 at the interface forming a hydrogen bond with the α-phosphate. In addition, T185 forms a hydrogen bond with the β-phosphate, while K181 contacts the γ-phosphate. In contrast, in the ATP-bound structure, the α-phosphate only forms a single hydrogen bond with the peptide backbone of G183, and R366 exclusively interacts with the γ-phosphate. Like pppGpp,

ADP is bound to G150D and WT ρ proteins via an extensive interaction network involving the α- and β-phosphates.

We also observed differences in the interaction networks at the interfaces between protomers depending on the nucleotide bound (Fig. 6b). Compared to WT/ATP, G150D/ADP exhibits more interactions along the entire protomer interface, involving the NTD (residues 1–120), the CTD (residues 154–419), and the intervening connector. Both G150D/ADP and WT/pppGpp exhibit slightly more contacts between their CTDs compared to WT/ATP, while in WT/pppGpp there are significantly fewer interactions along the NTDs and the connector regions (Fig. 6b). Altogether, our structural analysis suggests that when ρ is bound to ADP or pppGpp, the interaction network within the nucleotide-binding pocket is strengthened, in turn leading to changes at the protomer-protomer interfaces and enabling ρ to form an open hexamer. The nucleotide-specific alignment of the protomers likely accounts for changes in the geometry of the ρ hexamer that promote the formation of distinct higher oligomeric states observed in the presence of ADP vs (p)ppGpp.

In G150D filaments, an altered α5/α6 loop conformation is stabilized by a hydrogen bond between the D150 side chain and Y197 (Fig. 1e). Can the wild-type ρ adopt this state? Our modeling shows that reverse engineering of G150 into the G150D[F-ADP] structure does not alter the observed loop conformation. In ADP-bound open wild-type ρ hexamers, the loop is stabilized by hydrogen-bond interactions between the side chain of N151 and the side chain and backbone amide of S153. These interactions are broken in the filaments and would not be compensated by alternative contacts (between D150 and Y197) with the G residue at position 150. Instead, upon filamentation, the loop may be stabilized by conformational changes in the adjacent nucleotide-binding pocket (e.g., a hydrogen bond between the D156 side chain and the adenine N6, which we observe in the G150D filaments but not in the wild-type ρ-ADP structure; Fig. 6a). Importantly, however, although our modeling suggests that the wild-type ρ can adopt a filament-like α5/α6 loop conformation, we do not discern an obvious reason why these changes should be required for ρ to form filaments.

## Discussion

In this work, we show that disruption of a finely-tuned network of interactions that underpin the optimal function of the termination factor ρ can lead to its sequestration in inactive oligomers. Residue substitutions in the linker that connects ρ domains or binding of stress-associated nucleotides to the nearby ATPase site trigger the formation of dodecamers and higher-order polymers. ρ hexamers can form dynamic N-to-N dimers and long C-to-N filaments (Fig. 5f, g), which can be visualized by crosslinking in the presence of (p)ppGpp and ADP, respectively. Formation of stable (for cryoEM analyses) N-to-N dimers required prior crosslinking (Fig. 5f), and long ρ filaments could be directly observed on EM grids only if additionally stabilized by the C-terminal H[8] extension and linker substitutions (Fig. 1b, c). Double Cys substitutions at the protomer interface stabilized filaments formed by the untagged WT ρ, enabling their detection upon

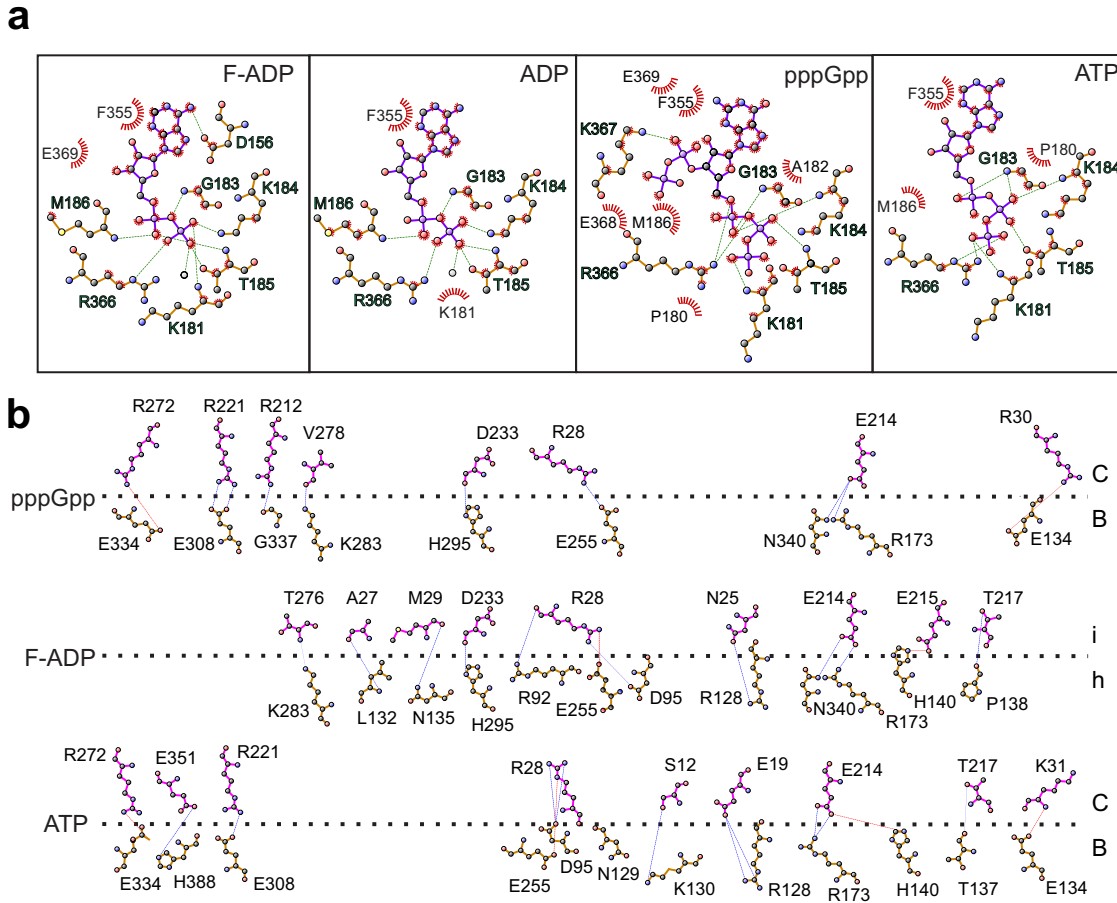

**Fig. 6 | ADP and (p)ppGpp promote hyper-oligomerization by changing ρ protomer interfaces. a** LigPlot representation of the nucleotide-binding modes of ρ. From left to right, G150D•H$_8$ filament bound to ADP (this study), untagged WT ρ bound to ADP (this study), untagged WT ρ bound to pppGpp (this study), and untagged WT ρ bound to ATP (PDB ID: 6WA8). Residues involved in binding to the indicated nucleotides are labeled. Green dashes, hydrogen bonds; red rays, van der Waals interactions. **b** DimPlot representation of interactions across selected protomer interfaces of pppGpp-bound untagged WT ρ (denoted pppGpp), ADP-bound G150D•H$_8$ filament (F-ADP), and ATP-bound untagged WT ρ (ATP; PDB ID: 6WA8) shown by a horizontal dashed line. For pppGpp- and ATP-bound ρ, interface contacts between subunits B and C are shown. For the ADP-bound G150D•H$_8$ filament, interface contacts between subunits h and i are shown. Subunits are identified on the right. Green dashes, hydrogen bonds; red dashes, salt bridges.

crosslinking; in contrast, substitutions that removed inter-protomer contacts destabilized ρ filaments (Supplementary Fig. 8).

We surmise that ρ filaments may serve as a storage form under low energy charge or stress, as indicated by elevated ADP or (p)ppGpp, and that nucleotide-stabilized inactive ρ oligomers must be intrinsically labile to ensure their facile reversal once growth resumes and ATP/ADP ratio rises; in agreement with this idea, ρ oligomerization is partially reversible in the presence of ATPγS in vitro (Fig. 5d). Unregulated filamentation, exhibited by the G150D mutant, leads to termination defects and sensitizes cells to stress (Fig. 4b and Supplementary Fig. 9). In the cell, ρ oligomerization is expected to be affected by intracellular crowding, interactions with protein partners and small molecules, and other cues. Although diverse proteins implicated in cellular metabolism have been shown to form filaments, the scarcity of methods that enable direct detection of native proteins within the complex intracellular environment poses a significant challenge to establishing the physiological relevance of filamentation[38]. Our findings that an eight-residue tag dramatically alters ρ oligomerization properties (Supplementary Fig. 6f) illustrate why a common approach, fusing a large fluorescent protein to the target to visualize aggregation in cells[38], may lead to artifacts. Future studies are necessary to determine the composition and structure of cellular ρ aggregates, reveal whether they represent transient storage depots or dead-end complexes targeted for disposal, and identify cellular factors that modulate ρ oligomerization.

Biomolecular phase separation covers a continuum of states, from dynamic liquid droplets held together by weak and transient interactions to highly ordered stable polymers[39,40]. Bacterial ρ proteins utilize this entire spectrum to control their activity and can form diverse aggregates. Intrinsically flexible open ρ rings can dimerize, stack into long filaments, or become stapled together during phage infection. We recently showed that multiple copies of the dimeric capsid protein of phage P4, Psu, can directly bind two open ρ rings, bridging them in the N-to-N configuration[41]. Interestingly, each of the individual ρ rings is stabilized by Psu in a helical geometry very similar to the conformation observed in the ADP-induced ρ polymers reported here, which enables ring expansion to at least the nonamer stage (see Supplementary Fig. 15 and ref. 41). Irrespectively, ρ hyper-oligomerization emerges as a versatile strategy to inactivate ρ in diverse stress situations.

We propose that unique features of ρ proteins underpin their different survival strategies (Fig. 7). While all ρ proteins share features that promote phase separation, e.g., multivalence and RNA binding[42], intrinsically disordered regions (IDRs), which mediate weak interactions in liquid droplets[43], appear to modulate properties of ρ across bacteria[44]. IDRs are rare in Bacillota and Pseudomonadota but common in Bacteriodota and Actinomycetota (Fig. 7a and Supplementary

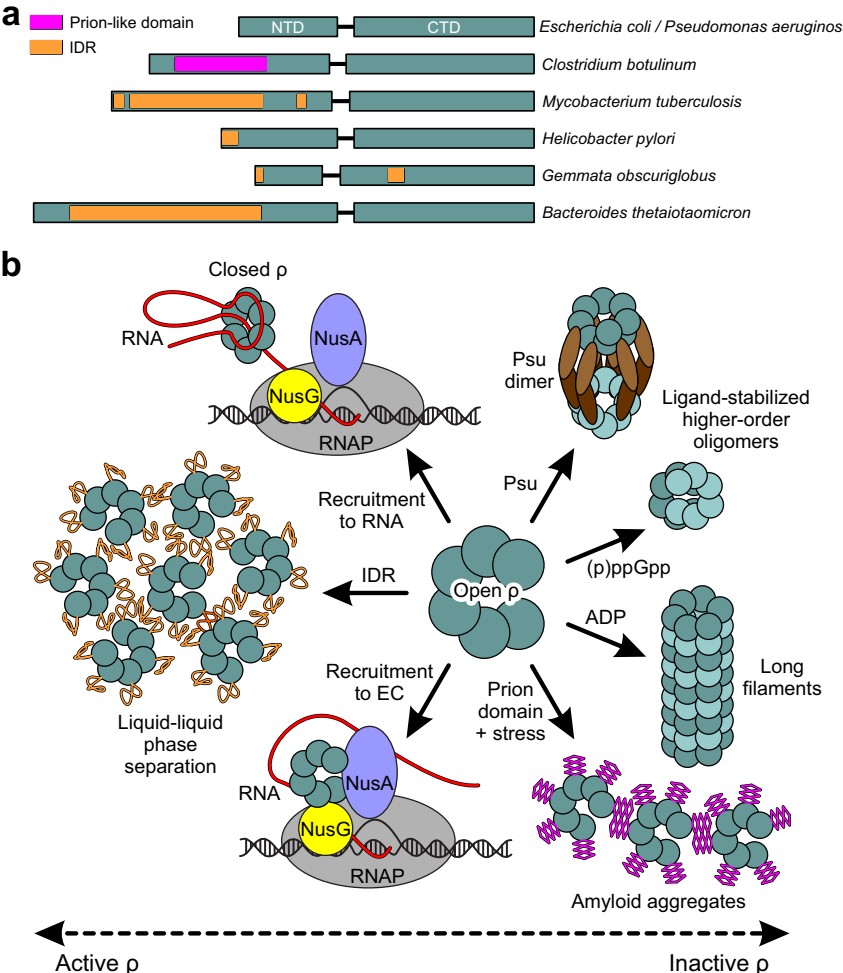

**Fig. 7 | Modulating ρ activity by oligomerization. a** Schematic diagrams of ρ proteins from representative bacteria. The IDR/prion-like domains with predicted local distance difference test (plDDT; see "Methods") < 50 modeled by AlphaFold 3 are shown as orange/magenta bars. **b** The ring conformation controls ρ activity; an open inactive hexamer can be stabilized by accessory proteins, such as Rof and YihE (not shown; see text for details) or trapped by ligand-induced polymerization/filamentation, as we observed with *E. coli* ρ. Formation of amyloids, whose structures remain to be determined, inactivates *C. botulinum* ρ, whereas phase separation increases ρ activity in *B. thetaiotaomicron*.

Fig. 16), and the residues at the protomer interfaces are variable (Supplementary Fig. 3). *Mycobacterium tuberculosis* IDR aids ρ binding to GC-rich sequences[45] and *Bacteroides thetaiotaomicron* IDR mediates the formation of liquid condensates that promote ρ activity and survival in the gut[46]. Conversely, in *C. botulinum*, a prion-like IDR fosters the formation of rigid amyloid aggregates in which ρ is inactive, a transition thought to promote adaptation to stress[47]. It remains to be determined if IDRs present in other bacteria control ρ activity. For example, *Helicobacter pylori* ρ has a short N-terminal IDR and *Gemmata obscuriglobus* ρ has short IDRs in both domains (Fig. 7a and Supplementary Fig. 16). Although IDRs known to alter ρ properties are large[45–47], our findings that the eight-residue terminal tag promotes oligomerization suggest that even short protein segments can have impacts.

Here, we show that *E. coli* and *P. aeruginosa* ρ proteins, which lack IDRs, form higher-order oligomers upon binding to ADP (Fig. 2a and Supplementary Fig. 7), a direct mechanism of sensing the adenylylate energy charge. The entrapment of ρ would serve to temper its termination activity and to prevent needless waste of ATP since ρ burns 1-2 ATP molecules per one traversed nucleotide[48]. A linkage between energy change and filament formation has been established for other enzymes, e.g., *E. coli* glutamine synthetase[49,50]. The reversible sequestration obviates a need for de novo protein synthesis when growth resumes, and stress/stasis-induced hibernation by multimerization is a common adaptive response shared by ribosomes, RNAPs, and metabolic enzymes in bacteria and eukaryotes[38,51–54]. In contrast, hyperoligomerization stabilized by Psu protein[41] is lethal in many pathogens[55], suggesting that ADP- and Psu-like ligands may be attractive drug leads.

## Methods

### Protein purification

Plasmids used for protein purification are listed in Supplementary Table 1. All of the restriction and modification enzymes for plasmid construction were from New England Biolabs. DNA oligonucleotides for vector construction and sequencing were obtained from Millipore Sigma, and synthetic DNA fragments for Gibson assembly were from IDT. The sequences of all plasmids were confirmed by Sanger sequencing at the Genomics Shared Resource Facility (Ohio State University). *E. coli* BL21 (DE3) cells were used for protein expression. Kanamycin (50 μg/ml) was added as needed.

Cells harboring ρ variants [all ρ variants are derivates of *E. coli* MG1655 ρ (NCBI ID: NP_418230.1 [https://www.ncbi.nlm.nih.gov/protein/NP_418230.1/]) unless stated otherwise] were cultured in Terrific Broth (TB) at 37 °C, and 0.5 mM isopropyl-1-thio-β-D-galactopyranoside (IPTG; Goldbio) was added when $OD_{600}$ reached ~0.5. After

incubating for another 2.5 h at 37 °C, the cells were collected by centrifugation at 8000 × $g$ for 8 min at 4 °C.

For His-tagged ρ variants, the cell pellet was resuspended in Lysis buffer A (25 mM Tris-HCl, pH 7.5, 5 % (v/v) glycerol, 1 M NaCl, 0.5 mM tris(2-carboxyethyl)phosphine (TCEP), 1 x ProBlock Gold 2D Protease Inhibitor Cocktail (EDTA Free; Goldbio)) and opened by sonication. The cell lysate was cleared by centrifugation (20,000 × $g$) for 30 min at 4 °C. The supernatant was passed through a 0.45 μm filter and applied to the HisTrap HP column (Cytiva). A linear gradient with Ni-A buffer (20 mM Tris-HCl, pH 7.5, 5% glycerol, 350 mM NaCl, 0.5 mM TCEP) and Ni-B buffer (Ni-A supplied with 1 M imidazole) were used for elution. The eluted protein was loaded onto the HiTrap Heparin HP column (Cytiva) and eluted with a linear gradient of NaCl (0 - 1 M) in Hep-A buffer (20 mM Tris pH 7.5, 5% glycerol, 0.5 mM TCEP).

For tag-free ρ, the cell pellet was resuspended in Lysis buffer B (50 mM Tris-HCl, pH 7.5, 10% glycerol, 100 mM KCl, 1 mM DTT, 1 x ProBlock Gold 2D Protease Inhibitor Cocktail (EDTA Free)). After sonication, cell lysate was cleared by centrifugation and loaded onto the HiTrap Heparin HP column. Protein was eluted as above and loaded onto Sephacryl S-300 HR (Cytiva), preequilibrated in 20 mM Tris pH 7.5, 5% glycerol, 0.5 mM TCEP, and 400 mM NaCl.

All ρ variants were dialyzed against dialysis buffer (10 mM Tris-HCl, pH 7.5, 5% glycerol, 500 mM NaCl, 0.5 mM TCEP) and flash frozen in liquid nitrogen. Concentrations of all variants represent ρ hexamer.

The ρ proteins used for structural analysis by single particle cryoEM were purified as described above with the following variations. ρ proteins were expressed in *E. coli* RIL (BL21) cells (Novagen) and a final size-exclusion purification step was incorporated. Here, ρ mutants or ρ WT were loaded to a Superdex 200 column (Cytiva) equilibrated in SEC buffer (10 mM Tris pH 7.5, 200 mM NaCl, 1 mM DTT). Fractions corresponding to the peaks were combined and concentrated. ρ proteins were flash-frozen in liquid nitrogen and stored at − 80 °C until further use.

Cells harboring *Pseudomonas aeruginosa* ρ (Pae-ρ) were cultured in TB at 37 °C, and 0.2 mM IPTG was added when $OD_{600}$ reached ~ 0.5 for overnight induction at 18 °C. The harvested cell pellet was resuspended in Lysis buffer C (40 mM Tris-HCl, pH 8, 5 % (v/v) glycerol, 0.5 M NaCl, 5 mM β-mercaptoethanol (β-ME), 1 x ProBlock Gold 2D Protease Inhibitor Cocktail (EDTA Free; Goldbio)) and opened by sonication. The cell lysate was cleared by centrifugation (20,000 × $g$) for 30 min at 4 °C. The supernatant was passed through a 0.45 μm filter and applied to the HisTrap HP column (Cytiva). A linear gradient with Ni-C buffer (40 mM Tris-HCl, pH 8, 5% glycerol, 300 mM NaCl, 5 mM β-ME, 0.1 mM phenylmethylsulfonyl fluoride (PMSF; Thermo Fisher)) and Ni-D buffer (Ni-A supplied with 1 M imidazole) were used for elution. The eluted protein was loaded onto the Resource Q column (Cytiva) after 1.5 times dilution with Q-A buffer (25 mM Tris pH 8, 5% glycerol, 5 mM β-ME). A linear gradient between Q-A and Q-B buffer (Q-A buffer plus 1 M NaCl) was applied. The fractions containing Pae-ρ were pooled and digested with ULP1 overnight at 4 °C. The next day, the sample was passed through HisPur Ni-NTA Resin (Thermo Fisher) to remove the His-SUMO tag and ULP1. The untagged Pae-ρ was loaded onto the HiTrap Heparin HP column and eluted with a linear gradient of NaCl (0 - 1 M) in a Q-A buffer. The purity of Pae-ρ is assessed by mass spectrometry.

## Sample preparation for cryoEM
Before grid preparation, all ρ variants were applied to a Superdex 200 Increase 3.2/300 column (Cytiva) equilibrated with 10 mM Tris pH 7.5, 200 mM NaCl, 2 mM DTT, and fractions of ρ were pooled and concentrated.

For ρ G150D and G152D, proteins (2.9 mg/ml) were incubated with 2.2 mM ADP-BeF for 5 min at room temperature. Subsequently, 3.8 μl of the ρ-ADP-BeF mix was applied to glow-discharged Quantifoil R1.2/ 1.3 holey carbon grids and plunged into liquid ethane using a Vitrobot

Mark IV (Thermo Fisher) set at 10 °C and 100 % humidity. For $\rho^{pppGpp}$ and $\rho^{ADP}$, WT ρ (7.8 mg/ml) was incubated with 10 mM pppGpp or ADP, respectively, for 5 min at room temperature. 3.8 μl of the ρ-nucleotide mix was applied to glow-discharged Quantifoil R1.2/1.3 holey carbon grids and plunged into liquid ethane using a Vitrobot Mark IV (Thermo Fisher) set at 10 °C and 100 % humidity. For BMOE crosslinked structure, $\rho^{X1}$ (0.6 mg/ml) in PN23 buffer (10 mM Tris-HCl, pH7.4, 0.5 mM TCEP, 12 mM MgCl$_2$, 230 mM NaCl) was incubated with 2 mM ppGpp for 10 min at room temperature. Then 0.5 mM BMOE was added, followed by 10 min incubation at room temperature in the dark. The reaction was quenched with 5 mM DTT for 5 min.

## CryoEM data acquisition and processing
CryoEM data were acquired on an FEI Titan Krios G3i TEM operated at 300 kV equipped with a Falcon 3EC direct electron detector. Movies were taken for 40.57 s, accumulating a total electron dose of ~ 40 e-/Å2 in counting mode distributed over 33 fractions at a nominal magnification of 96,000x giving a calibrated pixel size of 0.832 Å/px.

All image analysis steps were done with cryoSPARC[56]. Movie alignment was done with patch motion correction generating half-binned micrographs, followed by CTF estimation with Patch CTF. Micrographs for ρ mutants G150D and G152D unexpectedly revealed the filamentous organization of ρ besides hexamer particles. Class averages of manually selected particle images were used to generate an initial template for reference-based particle picking from 2511 micrographs for the ρ-G150D sample. 495,704 particle images were extracted with a box size of 192 px and Fourier-cropped to 96 px for initial analysis. Reference-free 2D classification was used to select 168,536 particle images of filamentous ρ for further analysis. Helix refinement applying a helical rise of 200 Å was used to generate an initial 3D reconstruction of the filament from a small subset of the data set. To obtain a better-resolved reconstruction, all selected particle images were refined against this reference low-pass filtered to 30 Å applying the non-uniform (NU) refinement routine. Local processing was allowed to start at a resolution of 8 Å. Particle images were re-extracted with a box size of 384 px after local motion correction and subjected to helix refinement giving a reconstruction at ~ 3.5 Å resolution. Another iteration of reference-free 2D classification was applied to select the best class averages, followed by local CTF refinement and helix refinement to generate the final reconstruction at ~ 3.3 Å resolution. Data for the ρ-G152D sample was analyzed accordingly, giving a final reconstruction at ~ 3.3 Å resolution.

Since WT ρ samples did not reveal any filamentous particles, ab initio reconstruction was used instead of helix refinement to generate an initial reference for 3D refinement by NU refinement. To account for the sub-stoichiometric composition of ρ oligomers and isolate hexamers, 3D variability analysis (3DVA) was conducted. Final data sets of 200,796 and 314,323 particle images were selected for NU refinement, giving reconstructions of wildtype ρ at 3.0 Å and 2.6 Å in the presence of pppGpp and ADP, respectively.

## Model building, refinement, and analysis
Coordinates of ρ (PDB ID: 6WA8 [open ring][57]) were docked into the cryoEM maps using Coot (version 0.8.9)[58]. ρ subunits were manually adjusted to fit the cryoEM density. Manual model building alternated with real space refinement in PHENIX (version 1.20.1)[59]. Data collection and refinement statistics are provided in Supplementary Table 2. Structure figures were prepared with ChimeraX (version 1.6.1)[60,61].

## In vitro pelleting assay
1 μM purified ρ in PN23 buffer was cleared at 10,000 × $g$ for 5 min at 20 °C immediately before pelleting assay. Nucleotides were mixed with the cleared ρ and incubated at room temperature for 10 min. Then 40 μL of reaction was centrifuged at 55,000 rpm (~ 110,000 × $g$) in a TLA100 rotor (Beckman Coulter) for 20 min at 20 °C. The supernatant was removed, and the pellet was soaked in 40 μL of RRB buffer (10 mM Tris-HCl, pH7.4,

1 mM DTT, 500 mM NaCl, 2 M Urea) for 1 h at room temperature before resuspending. The samples were analyzed on SurePAGE 4-12% gel (GenScript) and stained with GelCode Blue Safe Protein Stain (Thermo Fisher).

### In vivo pelleting assay

To determine the influence of translation-inhibiting antibiotics on ρ filament formation, *E. coli* Δ*rfaH rho*⁺ strain (IA228; all strains used in this study are listed in Supplementary Table 1) was cultured in LB at 37 °C, 250 rpm to $OD_{600}$ ~ 0.5. Then antibiotics were added at the following final concentrations: 30 µg/mL nalidixic acid, 100 µg/mL mupirocin, 50 µg/mL retapamulin, or 100 µg/mL rifampicin. Ethanol was added to 0.4% (v/v) as a negative control. To confirm the relevance of (p)ppGpp, 100 µg/mL mupirocin or 0.4% ethanol was applied to ppGpp⁰ (Δ*relA* Δ*spoT*) strain (gift from Christophe Herman, Baylor College of Medicine) at $OD_{600}$ ~ 0.5. After 30 min incubation at 37 °C with shaking, the cells were collected by centrifugation at $5000 \times g$ for 5 min at 20 °C. To further test the influence of stress removal on ρ filament, after 30 min exposure to mupirocin, 10 mL of cells were collected to serve as 0 min post-MUP sample (Fig. 4e). Another 10 mL of cells were washed three times with fresh LB without antibiotic, reinoculated into the 10 mL of fresh LB without antibiotic, and incubated at 37 °C with shaking. The cells were collected after 40 min recovery. The growth curves for mupirocin recovery were recorded with EPOCH 2 microplate reader (BioTek). To test the effects of cellular starvation of $Mg^{2+}$ on the formation of ρ polymers, we adapted a previously published protocol[31]. A modified MOPS medium (MM medium) was used: MOPS medium lacking $Mg^{2+}$ and $Ca^{2+}$ ions[62] was supplied with 0.1% casamino acids and 0.2% glucose. Strain IA228 was cultured overnight in MM medium supplied with 10 mM $MgCl_2$. The next day, cells were reinoculated into fresh MM medium plus 10 mM $MgCl_2$ and allowed to grow to $OD_{600}$ ~ 0.2 at 37 °C. Then cells were washed 3 times with MM medium to remove $Mg^{2+}$. Cells were resuspended in the same volume of fresh MM medium supplied with either 0 mM or 10 mM $MgCl_2$. Cells were pelleted after 2 and 3 h incubation.

Collected cells were opened using sonication in PN23 buffer supplied with 1 x ProBlock Gold 2D Protease Inhibitor Cocktail (EDTA Free; Goldbio). The cell lysate was centrifuged at $1000 \times g$ for 10 min at 4 °C, and the supernatant was passed through a 0.45 µm filter. Then 150 µL and 200 µL of cell-free cell lysate for wild-type strain and ppGpp⁰ strain, respectively, were centrifuged at 55,000 rpm in a TLA100 rotor for 30 min at 4 °C. The supernatant was removed, and the pellet was soaked in 30 µL of RRB buffer for 1 h before resuspending. 10 µL supernatant and 30 µL pellet were analyzed by Western blotting with anti-ρ polyclonal antibodies (a gift from Evgeny Nudler, New York University).

### Western blot analysis

Following separation in SurePAGE gels, the proteins were transferred to nitrocellulose membrane (Bio-Rad) by electrophoresis with 100 V for 1 h on ice in Transfer buffer (25 mM Tris, pH 8.3, 192 mM glycine, and 20% methanol). After the transfer step, the membrane was blocked in Blocking buffer (25 mM Tris-HCl, pH 7.4, 150 mM NaCl, 0.1% Tween-20, and 5% (w/v) Blotting Grade Blocker Non-Fat Dry Milk (Bio-Rad)) for 1 h at room temperature. The membrane was washed twice with TBST (25 mM Tris-HCl, pH 7.4, 150 mM NaCl, 0.1% Tween-20) followed by incubation with anti-ρ polyclonal antibodies (1:10,000) in Blocking buffer overnight at 4 °C. The next day, the membrane was washed five times with TBST and incubated with secondary antibody (1:10,000; Goat Anti-Rabbit IgG (H + L)-HRP Conjugate (Bio-Rad)) in Blocking buffer for 1 h at room temperature. The membrane was then washed five times with TBST. Finally, the membrane was incubated with Clarity Max Western ECL Substrate (Bio-Rad) and imaged with ChemiDoc XRS + System. Quantification was done with Image Lab v6.1 (Bio-Rad). The means and *p*-value were calculated in Excel (Microsoft).

### Ex vivo crosslinking

To visualize the filament formation in vivo, *E. coli* BL21 (DE3) cells harboring plasmids encoding His₈-tagged ρ variants were cultured in 50 mL of LB at 37 °C to $OD_{600}$ ~ 0.7 and protein expression was induced with 1 mM IPTG for a 2 h induction at 37 °C. Cells were pelleted by centrifugation and washed twice with 1 x PBS (10 mM phosphate buffer, 2.7 mM KCl, and 137 mM NaCl, pH 7.4). After resuspending in 1 x PBS, 1 mM bismaleimidoethane (BMOE; Thermo Fisher) was added. Crosslinking was done for 20 min at room temperature in the dark, followed by quenching with 280 mM β-ME for 10 min. The cells were collected and resuspended in sonication buffer (1 x PBS, 400 mM NaCl, 2 mM DTT, 0.1 mM PMSF, and 2 M Urea). After sonication, the cell lysate was cleared at $20,000 \times g$ for 10 min. Proteins were resolved on a SurePAGE 4-12% gel, stained with NTA-Atto 550 (Sigma), and visualized on Typhoon 5 (Cytiva) with Cy3 mode.

### In vitro crosslinking

Nucleotides were added at concentrations indicated in figure legends to 1 µM purified ρ^XI or G150^XI in PN23 buffer and incubated for 10 min at room temperature. Then 0.5 mM BMOE was added, followed by 10 min incubation at room temperature in the dark. The reaction was quenched with 280 mM β-ME for 5 min. To assay for filament dispersal, filaments were first formed by incubating 1 µM ρ^XI with 2 mM ADP/ppGpp in PN23 supplied with an additional 10 mM $MgCl_2$ for 10 min at room temperature. Then, 20 mM ATPγS was added, followed by 30 min incubation at room temperature. Then samples were crosslinked with BMOE and quenched as above.

To test the effect of RNA on the polymerization of ρ, RNA and 2 mM ppGpp/ADP were mixed with 1 µM ρ^XI in PN23 buffer. Two RNAs are tested: utrRNA corresponds to the first 138 bp of the 5′ end of the SARS-CoV-2 genome (NCBI ID: NC_045512.2 [https://www.ncbi.nlm.nih.gov/nuccore/NC_045512.2/]) and totalRNA is a total RNA extract from *E. coli* MG1655 strain. The utrRNA was tested at 10 µM, while the totalRNA was tested at 40 ng/µL. Then samples were crosslinked with BMOE and quenched as above.

To test the effect of ppGpp in physiological conditions, we assembled two physiological nucleotide mixtures according to a previous report[26]. The optimal NT, mimicking normal growth condition: 2.2 mM ATP, 0.43 mM ADP, 88 µM AMP, 1.1 mM GTP, 160 µM GDP, 58 µM GMP, and 40 µM ppGpp. The stringent NT, mimicking mupirocin stress: 2.365 mM ATP, 0.272 mM ADP, 57 µM AMP, 0.446 mM GTP, 70 µM GDP, 30 µM GMP, 1.163 mM ppGpp, and 126 µM pppGpp. The physiological nucleotide mixture was incubated with 1 µM ρ^XI in PN23 buffer for 10 min at room temperature. Then samples were crosslinked with BMOE and quenched as above.

After separation in SurePAGE 4–12% gels, His-tagged ρ was stained with NTA-ATTO 550, and tag-free ρ was stained with GelCode Blue Safe Protein Stain. Gels were scanned and quantified using Typhoon 5 and ImageQuant v5.2. The means and standard deviation (SD) were calculated in Excel (Microsoft).

### Sucrose gradients

Sucrose gradients were prepared as described in ref. 63. Briefly, different sucrose density layers were prepared in 5% sucrose steps, filtered through a 0.45 µm filter, and sequentially layered into 13.5 mL open-top thin-wall polypropylene tubes (Beckman Coulter), starting with the highest density; each layer was incubated at − 80 °C for 15–30 min before addition of the next.

For testing cell extracts, *E. coli* Δ*rfaH* strains with *rho*⁺ (IA228) and *rhoG150D* (IA305) chromosomal alleles were cultured in LB at 37 °C for 24 h. Cells were collected by centrifugation and opened using sonication in PN23 buffer supplied with 1 x Protease Inhibitor Cocktail. The cell lysate was centrifuged at $1000 \times g$ for 10 min at 4 °C, and the supernatant was passed through a 0.45 µm filter. Then, 100 µL of the cell-free lysate was loaded onto the sucrose gradient.

To analyze purified ρ, 1 μM ρ was incubated with indicated nucleotides for 10 min at room temperature. Then 100 μL of the reaction was loaded onto sucrose gradient. Ultracentrifugation of sucrose gradients was performed at $110,000 \times g$ in SW41Ti rotor (Beckman Coulter) for 16 h at 4 °C. Fractioning of the gradient was done by taking aliquots from the top of the gradient, and the pellet was resuspended in RRB buffer. The samples were analyzed by Western blotting unless stated otherwise.

### In vivo ppGpp sensitivity assays

Three assays were performed to compare the effect of (p)ppGpp on the WT and G150D *rho* strains. First, the *relA* deletion strains carrying either WT (IA793) or G150D (IA791) *rho* variants were transformed with plasmids expressing either WT or catalytically-dead D275G (p)ppGpp synthetase RelA from an IPTG-inducible promoter[25], plated on LB/Carb in presence of 20 μM IPTG, and incubated at 37 °C for 16 h (Supplementary Fig. 9b). Second, the effect of *relA* deletion on growth of WT and G150D strains was compared in liquid growth assays at 37 °C (Supplementary Fig. 9a). Third, strains with WT (IA228), G150D (IA305), and a *rho*-down [an IS2 insertion in *rhoL* (IA306)] alleles were streaked out on LB plates with or without 25 mg/L mupirocin and incubated overnight at 37 °C (Fig. 4b).

### Effects of filamentation on ρ ATPase and termination activity

The DNA template used in these assays was generated by PCR amplification with plasmid pIA267 and primers listed in Supplementary Table 1 and purified by a PCR cleanup kit (Qiagen).

In the ρ ATPase activity assay, primers Rut_UP and Rut_DN (Supplementary Table 1) were used to generate a DNA template. The *rut* RNA (a 221 bp ρ-utilization site containing RNA) was made by T7-RNAP-based in vitro transcription[64]. To test the effects of filamentation on ATPase activity, 1.5 μM ρ was incubated with 2.5 mM nucleotides (GDP/ADP/ppGpp) or water in RH buffer (10 mM Tris-HCl pH 7.5, 0.5 mM TCEP, 150 mM NaCl, 5 mM $MgCl_2$) for 10 min at room temperature to permit filament formation. Then *rut* RNA and ATP were added to 1.5 μM and 2 mM (containing 0.2 μCi/μL γ-$^{32}$P-ATP) final concentrations, respectively. After another 5 min incubation, the reaction was stopped by 70 mM EDTA. 1 μL of samples were analyzed by thin layer chromatography on PEI-cellulose plates (Millipore Sigma) in 1 M NaCl. The fraction of $P_i$ was quantified.

The DNA template for testing the effects of filamentation on ρ-dependent termination was produced by primers λPR_UP and λPR_DN (Supplementary Table 1). Two pre-mixtures were made: (1) ρ-nucleotide mix was formed by incubating 2.5 μM ρ with 5 mM nucleotides (GDP/ADP) or water for 10 min in RH buffer at room temperature; and (2) Halted A26 elongation complex was formed at 37 °C for 15 min by mixing 440 nM RNAP holoenzyme with 200 nM DNA template, 100 μM ApU, 10 μM GTP and UTP, 2 μM ATP and 0.4 μCi/μL α-$^{32}$P-ATP in TGA2 buffer (20 mM Tris-acetate, 20 mM Na-acetate, 2 mM Mg-acetate, 5% glycerol, 1 mM DTT, 0.1 mM EDTA, pH 7.9). Then the pre-incubated ρ-nucleotide mixture was diluted 5 times into the halted transcription A26 elongation complex. 150 μM NTPs and 25 μg/mL rifapentin were added to restart RNA synthesis. After 6 min at 37 °C, the reaction was stopped by the addition of an equal volume of STOP buffer (10 M urea, 60 mM EDTA, 45 mM Tris-borate; pH 8.3). Samples were heated for 2.5 min at 95 °C and separated by electrophoresis in denaturing 8 % urea-acrylamide (19:1) gels (7 M urea, 0.5 × TBE). The fraction of run-off (% RO) RNA was quantified.

The results were visualized and quantified using the FLA9000 Phosphorimaging System and ImageQuant Software, respectively.

### Analysis of hexamer geometry

For the calculation of hexamer geometry, as depicted in Fig. 1d, the "draw_rotation axis" script in PyMOL was utilized. The script calculates three values between two adjacent subunits: the rotation angle (β) with respect to the rotation axis, the length of the translation vector along the rotation axis (r, between A/B, B/C, C/D, D/E, and E/F), and p (between A/F). In addition, it computes the distance between two adjacent subunits (designated as "g" for the A-F distance). Subsequently, these values were employed to calculate the upward rotation (α) between subunits. The presented values for α and r represent the means determined between A/B, B/C, C/D, D/E, and E/F.

### ρ conservation

To compile a representative database for ρ conservation study, the GTDB[65] bacterial taxonomy list (Released April 08, 2022) was downloaded. Genomes labeled as "gtdb_type_species_of_genus" and "gtdb_representative" were kept. Only complete genomes were used to build the database. The representative dataset covers 42 Phyla including 643 Genera. The protein sequences of all genomes were downloaded from NCBI.

There are three Pfam model hits for *E. coli* ρ, PF00006 (ATPase domain; residues 158–365 of *E. coli* ρ), PF07497 (RNA-binding domain; residues 51–125), and PF07498 (N-terminal domain; residues 5–47). The PF07498 model encompasses a part of the N-terminal domain that lacks key functional residues of ρ, whereas PF00006 represents many very diverse nucleotide-binding proteins. In addition to these domain-specific models, an NCBI HMM model of full-length ρ (NF006886.1 [https://www.ncbi.nlm.nih.gov/protfam/?term=NF006886.1]) is available. However, noting that a model based on a full-length protein could miss some novel sequences, we first used PF07497 to identify as many ρ candidate sequences as possible by searching against the representative database with hmmsearch (v.3.3)[66] with a bit score of 27, a suggested gathering threshold. The identified ρ-like proteins were examined using NF006886.1 with the suggested cutoff of 315. If a sequence failed to pass this cutoff, we applied two additional filters to determine if it could be a real ρ: (1) PfamScan (https://www.ebi.ac.uk/jdispatcher/pfa/pfamscan) to check the domain architectures and (2) a reciprocal blast search (blastp, BLAST + v.2.9.0)[67] with ρ from *E. coli* MG1655 (NCBI genome ID: GCF_000005845.2 [https://www.ncbi.nlm.nih.gov/datasets/genome/GCF_000005845.2/]) to identify true orthologs, which tend to find each other as the reciprocal best hit. Two sequences, WP_236262298.1 and WP_146920178.1, which lack convincing hits for the ATPase domain and are not the reciprocal best hits with *E. coli* ρ, were discarded.

The final dataset contains 632 ρ sequences (Supplementary Data 1a) that range from 355 to 857 residues in length. Multiple sequence alignment (MSA) was done by Dialign v.2.2.1[68] with a default setting, which enables detection of local homologies in sequences with low overall similarity, and indexed according to *E. coli* ρ (NP_418230.1). The MSA was inspected manually in Jalview v.2.11.2.7[69]. The sequence logo was generated using WebLogo (v.3.7.8)[70].

### ρ IDR distribution analysis

Intrinsically disordered regions (IDR) and prion-like domains were identified (Supplementary Data 1b) by metapredict v.2.6[71] and PrionW[72], respectively. The representatives shown in Fig. 7a were submitted to AlphaFold 3[73] for calculating the predicted local distance difference test (plDDT). A very low confidence score is an indication of a disordered region[74]. To visualize the IDR-containing ρ distribution in each phylum, a ρ sequence similarity network was built. An all-vs-all blastp was performed using BLAST + (v.2.9.0)[67] with an E-value of $10^{-10}$, and the result was explored in Cytoscape (v.3.9.1)[75]. The percentage identity (%id) was used to group ρ sequences. Our analysis showed that %id > 75 can separate ρ from γ- and α-Proteobacteria, which have the highest fraction of ρ representatives (Supplementary Data 1a). Thus, ρ sequences are grouped on the "class" level. The pairwise alignments of the all-vs-all blastp results reveal that more than 85% of the connected pairs in the similarity network show 80 +% query coverage and that

lower query coverage is due to the presence of IDRs. Consistent with recent reports,[44,46] the IDRs show remarkable sequence diversity.

## Reporting summary

Further information on research design is available in the Nature Portfolio Reporting Summary linked to this article.

## Data availability

CryoEM reconstructions have been deposited in the Electron Microscopy Data Bank [https://www.ebi.ac.uk/pdbe/emdb] under accession codes EMD-18132 ($\rho^{G150D}$), EMD-18133 ($\rho^{G152D}$), EMD-18131 ($\rho^{pppGpp}$), EMD-18130 ($\rho^{ADP}$) and EMD-50352 ($\rho^{X1-BMOE}$). Structure coordinates have been deposited in the RCSB Protein Data Bank [https://www.rcsb.org] with accession codes 8Q3P ($\rho^{G150D}$), 8Q3Q ($\rho^{G152D}$), 8Q3O ($\rho^{pppGpp}$), 8Q3N ($\rho^{ADP}$) and 9FF7 ($\rho^{X1-BMOE}$). Structure coordinates used in this study are available from the RCSB Protein Data Bank [https://www.rcsb.org] under accession codes 6WA8 and 8PEW. This paper does not report the original code. All other data are contained in the manuscript or the Supplementary Information. Source data are provided in this paper.

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

## Acknowledgements

We thank Georgy Belogurov, Lydia Freddolino, Jan Löwe, and Natacha Ruiz for the discussions, Evgeny Nudler for anti-ρ antibodies, Cristophe Herman for the ppGpp$^0$ strain, and Mike Laub for the RelA-expression plasmids. This work was supported by grants from the National Institutes of Health (GM067153 to I.A.), the Deutsche Forschungsgemeinschaft (INST 130/1064-1 FUGG to Freie Universität Berlin; WA 1126/11-1, project number 433623608, to M.C.W.), and the Berlin University Alliance (501_BIS-CryoFac to M.C.W.). We acknowledge the assistance of the core facility BioSupraMol supported by the Deutsche Forschungsgemeinschaft in electron microscopic analyses and the Sanger Sequencing at the OSU Comprehensive Cancer Center supported in part by NCI P30 CA0168058.

## Author contributions

B.W. analyzed filament formation using in vitro and in vivo approaches and performed bioinformatic analyses. N.S. assembled complexes for structural analysis and built atomic models. T.H. acquired, processed, and refined cryoEM data. M.F. performed growth assays. I.A. constructed plasmids for protein expression. I.A. wrote the first draft with contributions from N.S. and B.W. I.A. and M.C.W. coordinated the

project. All authors contributed to data interpretation and manuscript revisions.

## Funding

## Competing interests

The authors declare no competing interests.
