## [Peer Review file · Nature Communications]

Nucleotide-induced hyper-oligomerization inactivates transcription termination factor ρ

Corresponding Author: Professor Irina Artsimovitch

Version 1:

Reviewer comments:

Reviewer #1

(Remarks to the Author)

Rho, an ATP-dependent helicase, binds to ssRNA and/or RNA polymerase (RNAP), translocates along the RNA, and applies mechanical force to RNAP to trigger transcription termination. Under permissive growth conditions, tight coupling of transcription and translation prevents Rho from accessing its substrate RNA/RNAP due to steric hindrance by the ribosome. The manuscript explores how Rho maintains specificity under cellular stress, where inefficient translation might allow lethal excessive Rho termination. The manuscript posits that oligomerization is crucial, a conclusion initially unconvincing due to incomplete data in the original manuscript. Unfortunately, the conclusion is still not sufficiently supported by the data. Although the G150D protein forms filaments, there is no evidence that the WT protein forms the same filaments. In the last review, we suggested making mutations that disrupt the filament interface to test if they alter the pelleting of WT Rho. We also add here that biological confirmation would be that these mutations might rescue the G150D phenotype in the presence of MUP (Fig. 4). In the absence of structure-guided function experiments, we are not convinced that the filaments observed in the G150D structures occur with WT Rho, and we cannot recommend acceptance in Nat. Comm. without mutational studies. Below is a more in-depth review.

The new manuscript's main finding reveals that under specific conditions (+ADP, with a C-terminal His tag), mutant Rho proteins can form extensive filaments visible via electron microscopy. Single-particle cryoEM analysis shows these filaments locking G150D Rho in conformation, likely rendering it inactive. In addition, in the revised manuscript, Rho mutants crosslinked with pppGpp reveals a novel N-to-N structure that is distinct from the C-to-N filament structure observed, consistent with the distinct biochemical properties of Rho incubated with pppGpp versus ADP.

While no extended filaments were seen under other conditions, further biochemical and genetic work supports that wild-type (WT) Rho can reversibly oligomerize under specific conditions. Sensor cysteines capable of crosslinking only in the filament conformation help validate the cryo-EM structure's relevance. Additionally, the consistent nucleotide-dependence of Rho's pelleting and oligomerization supports dynamic oligomerization as a functional switch during cellular stress.

However, the His tag data seem extraneous to the biological findings and could confuse general readers of Nature Communications. We noted this in our earlier review but the authors argue that this artifact is important because others in the field use His-tag protein. We thus request that any biochemical data showing His-tagged variants be moved to the supplement, while the main figures should only show data from non-His-tagged protein. We believe this will allow readers to focus on the biological relevance of the findings in the main text, while permitting specialists in the field to dig deeper into His-induced changes in the supplement. Currently, WT and His-tagged data are intermingled in the main figures such that it becomes confusing to understand what the biologically-relevant versus potentially artifactual protein concentrations are.

The figures also require improved labeling and legends. Notably, each panel displaying a model should clearly indicate the protein and observed ligand, including source references.

Recommendations for Revision:

1. Label figures 1-4 and clarify legends to enhance comprehensibility.
2. Omit or move His tag data to supplement to streamline the narrative. Main figures should speak directly to biological relevance, rather than technical details of interest to specialists.
3. Detail the genetic and biochemical methodologies supporting reversible oligomerization and its physiological relevance.

Specific Figure Comments:

1. Figure 1:

- o In the accompanying text and Figure Title, please incorporate sooner that a His8 tag is required to observe the filaments.
- o (b) Please include the same WT-Rho experiment for comparison to demonstrate that G150D/G152D are also required for filamentation, not just the His tag.
- o (c) Unclear contents; please specify map versus model in the figure and legend. In addition, we recommend that a locally filtered map be provided in Fig. 1C to better support the sections of the model built at the periphery.
- o (e), (f), (g) Specify which structures are depicted, and show corresponding maps for H-bond claims. Some of these structures are not from this paper, right? If so, are they crystal, cryoEM, same buffers and concentrations of ligands?

2. Figure 2:

- o (a) Remove distracting His-tag data after noting it in the begin and consider a supplementary figure or appendix for it. Is there un-tagged G150D data that could be provided for comparison with WT instead?
- o (b) Can't clearly see the text that is overlaid onto the figure
- o (d) This table is difficult to digest; bar graph visualizations would be more helpful. In addition, are these with tagged or untagged proteins? No His tag is indicated, yet the ADP concentration used was 2 mM (while in the legend for 2A, it is said that you used 2.5 mM ADP for untagged Rho).
- o (e), (f) Move to supplement for specialists who want clarity on the His tag effects

3. Figure 3:

- o Can experimental density be provided for the previously published WT-ATP structure in to support their claims in Fig. 3A?
- o Why the different colors in a? The use of the same colors between 3a and 3c makes it seem as if the schemes are related, but they are not. It would be more clear to provide text labels at the top of the figure with the specific Rho allele and ligand written next to one another.
- o This figure shows that ADP pre-incubation inhibits Rho activity. These are nice data. However, raw gel images were not provided for Fig. 3b and Fig. 3c in the accompanying Excel file (we do appreciate seeing the replicates for the other crosslinking and sucrose gradient experiments). Please provide these.
- o Please specify how the p-values were calculated in the Figure Legend.

4. Figure 4:

- o Clarify the experiment and what the y-axes represents in Fig. 4 c, d and e. It is not clear that these are in vivo pelleting assays, rather than sucrose gradient fractionation experiments as in Fig. 4a.

5. Figure 5:

- o Where did the structure in Fig. 5c come from? Please provide PDB ID if previously-published.

6. Figure 6:

- o PDB ID missing in Figure Legend

Text:

- Lines 122-125: "In contrast, in the ATP-bound state, the R353-W381 contact is more dynamic, as we observed distinct orientations of these two residues within the hexamer. The cation- π interaction must be disrupted during ring closure, and higher flexibility of R353 and W381 likely facilitates the underlying conformational changes." The argument that R353 and W381 are more dynamic/flexible when ATP is bound is currently not supported. The rightmost panel of Fig. 3A implies that R353 and W381 sample one conformation in the ATP-bound state (albeit a different conformation than the ADP-bound states), while the text implies that multiple conformers can exist when ATP is bound. Please clarify this point.

PDBs

- Please revise the title of PDB entries to clearly indicate whether Rho is His-tagged; whether Rho is mutated (for instance, in the BMOE structure, presumably the mutant cysteine is present?) and specify bound ligands (e.g. does the G152D structure also contain ADP?)

Reviewer #2

(Remarks to the Author)

Reviewer #3

(Remarks to the Author)

Wang et al. have substantially improved their manuscript upon incorporating new data, reorganizing the manuscript structure, and tuning down the most hypothetical claims. What is clear from the work is that E. coli's Rho is able to form higher-order, inactive oligomers upon the introduction of a G-to-D substitution in the connector loop between Rho's NTD and CTD or upon binding to ADP or ppGpp. The authors provide cryoEM structures of both Rho filaments and Rho N-to-N

double-ring dodecamers that illustrate the versatility of Rho hyper-oligomerization in vitro (or ex vivo under conditions of Rho overexpression from a plasmid). The hyper-oligomers appear to be rather unstable and require either a His8-tag or chemical cross-linking to be observed by cryoEM. Under conditions of translational or transcriptional stress, the authors find Rho in ultracentrifugation pellets from cell lysates and tentatively connect these aggregates to the hyper-oligomers observed in vitro. Despite their efforts, the formation of the hyper-oligomers under physiological conditions remains uncertain and further (likely difficult) work will be required to rigorously establish this point and rule out alternative scenarios. Nevertheless, I think that the authors provide an extensive and intriguing body of work that challenges our current conception of Rho-dependent regulation, and that will likely prompt follow-up studies from groups interested by this topic.

Specific points:

- 1) Define 'rut' first time used (lane 31).
- 2) Remove mention of the RNAP-centered scenario as the favored scenario (lanes 35-36). This point is hotly debated and, in any case, favoring this or other scenarios has absolutely no consequence for the present work.
- 3) The description of the structural details on pages 4-5 is difficult to follow. I suggest that the authors provide positions of the various loops/helices they discuss, either on the diagram of fig1a (where only helix alpha 6 is depicted) or in fig S3a.
- 4) More generally, I think the clarity of the manuscript would benefit from mentioning more frequently the figure panels where the readers can find the reported findings.
- 5) The in vitro transcription termination data now reported in figure 3c provide an important control BUT: i) it is customary to provide a representative gel image that will help experts assess the nature of the effects. For instance here, how the inhibitors affect the distribution of signals between the various termination sites, whether the overall transcription yield is also affected, etc. ii) why has ppGpp not been tested alongside the other nucleotides?
- 6) The statistical test(s) used in Fig 3b,c is(are) not mentioned.
- 7) Lane 167: "...since (p)ppGpp binding alone is not sufficient to promote pelleting...". Where is the supporting data in the current manuscript?
- 8) Lanes 208-214: The authors use sucrose gradient as an independent means to detect formation of oligomers with native Rho (instead of the Cysteine-modified variants used in BMOE cross-linking assays). There is a strong pellet signal, suggesting oligomerization in presence of ADP (figS12b), but the protein used is Rho-His8, not native Rho. Is the same signal observed with native (untagged) Rho? Again, this questions the relevance of the authors' findings as potential lab curiosities.

Reviewer #4

(Remarks to the Author)

Review of "Transcription termination factor ρ polymerizes under stress" Wang et al. 2024.

As I wrote in my initial review, I have an overall positive opinion of this manuscript, which I think is suitable for publication in a high quality, broad scope journal.

My technical expertise is in bioinformatics, so I will almost limit my review to this aspect (comparative sequence analysis), which is definitely not the most important of the manuscript.

In their rebuttal and revised manuscript, the authors have attempted to respond to the comments I have made. In particular, I thank the authors for clarifying some points of their approach in the rebuttal, such as the use of "decoy sequences" in the construction of the similarity network. Some of my suggestions have also been implemented in the MS (in the direction of simplifying the analysis).

Unfortunately, I still think that some aspects of this part of the MS need some work. To be honest, I find text L497-544, describing the approach, very difficult to understand. I feel that it has not been written with the same attention to detail as the rest of the manuscript and I do not think it meets the standards for a publication of this level, I mention a number of specific points below but I think it needs extensive rewriting. I also raise a specific concern about the approach used to compute the similarity network (which does not invalidate the results, but makes the workflow somewhat more cumbersome than necessary), see below.

Specific comments :

- * L503 Avoid the abbreviated description for the three PFAM models (ATP-synt_ab, Rho_RNA_bind, Rho_N) or introduce them correctly.
- * L504 The sentence "PF00006 and PF07498 find too many sequences..." seems incorrect. For example "Searching (hmmsearch, v3.3) with PF00006 and PF07498..." would be more correct.
- * L506 "especially when the sequencing data accumulates faster than ever" is not necessary.
- * L513 Why the future tense "will examine"?
- * L514 and L518 The contractions "didn't" and "don't" sound inappropriate in formal academic writing and are not used elsewhere in the text.

* L516 "reciprocal blast" is incorrect in formal academic writing (perhaps replace with "reciprocal blast search"). The tool should also be introduced here (blastp, BLAST+ v 2.9.0) and not later (currently L534).

* L516 I think the accession number number for the genome sequence would be more appropriate than taxid for *E. coli* MG1655 in this context.

* L525 I do not understand the use of different tenses in adjacent sentences ("The MSA is inspected ...", "Sequence logo was ...").

* The term "sequence logo" should probably be mentioned in the legend of Figure 2E to help the reader connect the methods described with the results.

* L256 and section "p IDR distribution analysis". Having read the rebuttal carefully, I now understand better the use of "decoy sequences", which are only used here to select an E-value cut-off, not to remove outliers. However, I do not think that this terminology is common in this context (in contrast, the term seems to be relatively common in structural biology...). Furthermore, I still do not understand why the authors need to choose a cut-off on the E-value: if the set of 632 p sequences is correct in the sense that it contains no outliers (i.e. only p sequences), the authors should no longer be concerned about statistical significance. They can simply do a pairwise sequence alignment and apply a 75% id cut-off (probably only after also applying a cut-off to the length of the alignment, otherwise the id percentage may be high but the alignment may only cover a fraction of the protein). Alternatively, the authors may not re-align with blast and simply compute the pairwise sequence similarity on the MSA produced by L521-523?

* Figure 7b can be clarified by indicating representative species for the different modulation mechanisms.

* L257 This is a personal opinion, but I do not like the sentence "We propose that phylum-specific features of p proteins underpin their different survival strategies". To me, the relationship between the variety of mechanisms implemented to modulate p activities discussed in the MS and the survival strategies is unclear (specific strategies may simply be implemented with the modulation mechanism available in the organism under consideration).

Version 2:

Reviewer comments:

Reviewer #1

(Remarks to the Author)

The points raised in my previous report have been satisfactorily addressed.

Reviewer #5

(Remarks to the Author)

Review NCOMMS-23-60851B

Wang et al have satisfactorily addressed my concerns in their revised manuscript. The manuscript would further benefit from the following minor changes:

1) More consistently indicating which version (H8 or untagged) of Rho was used in each of the figure panels. This is not always obvious without digging deep in the paper and even sometimes remains uncertain.

2) Indicating the degree of conservation between Rho factors from *E. coli* and *P. aeruginosa* (~81% identity and ~89% similarity if I am not mistaken), e.g. in lane 93. Figure 7a could also be modified to indicate that the top schematic is for both *E. coli* and *P. aeruginosa* Rho factors.

I congratulate the authors for their extensive, somewhat provocative work and for their efforts at improving their manuscript during the revision stages. I am eager to see if future work will indeed prove unambiguously the physiological relevance of their "Rho polymerization under stress" model.

Reviewer #6

(Remarks to the Author)

Many hexameric ATPases (in particular AAA+ ATPases) are known to have a tendency to form higher order oligomers and filaments although it has been shown that the active form is a non-planar hexamer. It has been proposed in the past that the inactive higher (than hexamer) order oligomers could be a storage form although this was not observed in cells. Given the lack of functional activities in vitro and inability to observe in cells, the idea of storage form could not be adequately tested. Furthermore, most of the higher order oligomers were observed in vitro with mutants or with high concentration of one particular type of nucleotide (such as ADP or AMPPNP), making it more difficult to argue this is physiologically relevant in cells, where there will be a mix of ATP and ADP.

The manuscript here reports in vitro and in vivo data on Rho hexamer helicase, arguing that the filaments and dodecamers they observe in vitro (with mutants or ADP or pppGpp) could represent inactive forms in cells. However, these fall into similar conundrum as data on higher order oligomers shown previously. It is plausible but difficult to prove. The manuscript presents

various in vitro and in vivo data. But the data are sometimes inconsistent, partly due to the varying effects of his-tags, mutations, nucleotides and in vitro vs. in vivo. The authors make little attempt to provide/reconcile mechanistic/molecular basis for the data. For example, there are multiple paragraphs/subsections on the mutant G150D, and the requirement of ADP/ATP on filaments/oligomer formations. ADP is required for G150D filaments formation in cells as in their pelleting assays, but not required in vitro. In the structures, they show the importance of bound nucleotide in stabilising the filaments, so one would expect ADP does promote filaments formation in vitro. The authors should try to resolve the discrepancy on whether nucleotides (either ATP or ADP) promote filament formation. Could it be due to higher protein concentrations in in vitro experiments? This could be tested the in vitro effects with varying protein-concentrations. Furthermore, they seem to argue ADP stabilises open ring conformation, rather than filaments, causing further confusions.

Overall, it is important to demonstrate the filaments can form for WT proteins. Even if the filaments are unstable, the authors should try crosslinking (with their cysteine mutants) before putting onto EM grids (either negative stain or cryo). Direct visualisation is important even though the numbers of filaments might be low and from which it will be difficult to obtain high resolution structures. Ideally its relevance in vivo should be demonstrated. Based on the structures, are there potential mutations that could interfere with higher-order oligomer formation without interfering with active open ring formation ? If so, introducing these mutations in vivo might demonstrate the importance of these storage/inactive forms in cellular activities.

The data on wildtype Rho aggregation in cells under stress do not necessarily support that they form filaments or higher order oligomers, as there are so many processes that have been affected in these cells (translation, transcription and others affected by antibiotics). Since Rho interacts with many other proteins and RNA, which can also aggregate under stress, the observed Rho aggregation could therefore be indirect.

Minor comments:

Why did ppGpp structure show open ring conformation even though it showed similar levels of dodecamer formation as with pppGpp? It is unclear what the effects of ppGpp or pppGpp are.

Since G150D induces major conformational changes in the loop, would this be possible in WT proteins ? If this is possible, how would stress induce the changes that stabilise the conformation ?

If filaments exist in cells. How are the stable filaments converted back to active hexamers ?

Response to Reviewer Comments

Reviewer comments are repeated in **bold**, our responses are in *italics*. Significant changes are indicated in **red** in the revised manuscript; only a few key textual edits are included in the rebuttal.

Review of “Transcription termination factor ρ polymerizes under stress” Wang et al. 2024.

As I wrote in my initial review, I have an overall positive opinion of this manuscript, which I think is suitable for publication in a high quality, broad scope journal.

My technical expertise is in bioinformatics, so I will almost limit my review to this aspect (comparative sequence analysis), which is definitely not the most important of the manuscript.

In their rebuttal and revised manuscript, the authors have attempted to respond to the comments I have made. In particular, I thank the authors for clarifying some points of their approach in the rebuttal, such as the use of “decoy sequences” in the construction of the similarity network. Some of my suggestions have also been implemented in the MS (in the direction of simplifying the analysis).

Unfortunately, I still think that some aspects of this part of the MS need some work. To be honest, I find text L497-544, describing the approach, very difficult to understand. I feel that it has not been written with the same attention to detail as the rest of the manuscript and I do not think it meets the standards for a publication of this level, I mention a number of specific points below but I think it needs extensive rewriting. I also raise a specific concern about the approach used to compute the similarity network (which does not invalidate the results, but makes the workflow somewhat more cumbersome than necessary), see below.

We thank the reviewer for their positive assessment of our work. We revised the manuscript following the reviewer’s suggestions to address all specific points and to streamline the description of the bioinformatics approach. We hope that we have succeeded in making this part of the manuscript more understandable.

Specific comments :

*** L503 Avoid the abbreviated description for the three PFAM models (ATP-synt_ab, Rho_RNA_bind, Rho_N) or introduce them correctly.**

We have updated descriptions for the PFAM models, included Rho residues represented by these models, and explained their limitations. The revised text states:

PF00006 (ATPase domain; residues 158-365 of *E. coli* ρ), PF07497 (RNA-binding domain; residues 51-125), and PF07498 (N-terminal domain; residues 5-47). The PF07498 model encompasses a part of the N-terminal domain that lacks key functional residues of ρ , whereas PF00006 represents many very diverse nucleotide-binding proteins. In addition to these domain-specific models, an NCBI HMM model of full-length ρ (NF006886.1) is available.

*** L504 The sentence “PF00006 and PF07498 find too many sequences...” seems incorrect. For example “Searching (hmmsearch, v3.3) with PF00006 and PF07498...” would be more correct.**

This sentence was deleted.

*** L506 “especially when the sequencing data accumulates faster than ever” is not necessary.**

Deleted.

*** L513 Why the future tense “will examine”?**

Revised.

* L514 and L518 The contractions “didn’t” and “dонт’t” sound inappropriate in formal academic writing and are not used elsewhere in the text.

Changed to “If a sequence failed to pass this cutoff...” and “...lack convincing hits...”

* L516 “reciprocal blast” is incorrect in formal academic writing (perhaps replace with “reciprocal blast search”). The tool should also be introduced here (blastp, BLAST+ v 2.9.0) and not later (currently L534).

Replaced with “reciprocal blast search” and introduced BLAST+.

* L516 I think the accession number number for the genome sequence would be more appropriate than taxid for E. coli MG1655 in this context.

The genome accession number GCF_000005845.2 is used now.

* L525 I do not understand the use of different tenses in adjacent sentences (“The MSA is inspected ...”, “Sequence logo was ...”).

These sentences were revised to say “The MSA was inspected manually in Jalview v.2.11.2.7⁶⁸. Sequence logo was generated using WebLogo (v.3.7.8)⁶⁹”.

* The term “sequence logo” should probably be mentioned in the legend of Figure 2E to help the reader connect the methods described with the results.

The legend of Figure 2e (now Supplementary Fig. 6a) was changed to “Sequence logo showing the conservation...”

* L256 and section “p IDR distribution analysis”. Having read the rebuttal carefully, I now understand better the use of “decoy sequences”, which are only used here to select an E-value cut-off, not to remove outliers. However, I do not think that this terminology is common in this context (in contrast, the term seems to be relatively common in structural biology...). Furthermore, I still do not understand why the authors need to choose a cut-off on the E-value: if the set of 632 p sequences is correct in the sense that it contains no outliers (i.e. only p sequences), the authors should no longer be concerned about statistical significance. They can simply do a pairwise sequence alignment and apply a 75% id cut-off (probably only after also applying a cut-off to the length of the alignment, otherwise the id percentage may be high but the alignment may only cover a fraction of the protein). Alternatively, the authors may not re-align with blast and simply compute the pairwise sequence similarity on the MSA produced by L521-523?

We agree with the reviewer that all 632 p sequences from the final dataset are true homologs. The “decoy sequences” helped us to set a correct E-value for the network analysis. We used decoy sequences to ensure the E-value was significant enough to separate true homologs from similar sequences, but not too-stringent to avoid losing some true connections among p sequences. Exploring the E-value to separate decoy sequences from p sequences in the network showed that we can use as low as 10^{-30} in the all-vs-all blastp search. But we also noticed that, after applying 75% id cut-off, the E-values among p sequence pairs are on the level of 10^{-140} , eliminating concerns about using the too-stringent cut-off. Thus, to simplify the description, we removed the usage of “decoy sequences” from the manuscript and their accession numbers from Dataset 1. In terms of query coverage, we have examined the pairwise alignments of the all-vs-all blastp result and added the following statements:

The pairwise alignments of the all-vs-all blastp results reveal that more than 85% of the connected pairs in the similarity network show 80+% query coverage and that lower query coverage is due to the presence of IDRs.

Consistent with recent reports,^{43,45} the IDRs show remarkable sequence diversity.

* Figure 7b can be clarified by indicating representative species for the different modulation mechanisms.

The Figure 7b legend was updated to indicate the representative species:

...ligand-induced polymerization/filamentation, as we observed with *E. coli* ρ . Formation of amyloids, whose structures remain to be determined, inactivates *C. botulinum* ρ , whereas phase separation increases ρ activity in *B. thetaiotaomicron*.

* L257 This is a personal opinion, but I do not like the sentence "We propose that phylum-specific features of ρ proteins underpin their different survival strategies". To me, the relationship between the variety of mechanisms implemented to modulate ρ activities discussed in the MS and the survival strategies is unclear (specific strategies may simply be implemented with the modulation mechanism available in the organism under consideration).

While the discussed examples come from a single species, we posit that the regulatory strategy are likely broadly specific. We observed that the absence/presence of IDR/prion-like domains is uniform inside a phylum, at least on the class level. Our results agree with a recent report (PMID: 38966992) which shows that a phylum tends to have the same ρ types. Rho is a global and essential (in most tested bacteria) regulator, and we posit that its regulatory strategies, which are at least in part determined by its sequence features, are likely shared among related bacteria. This model does not exclude the possibility that Rho may be controlled differently in an organism that inhabits an exotic ecological niche. We replaced "phylum-specific" with "unique" in the sentence in question.

Reviewer #3 (Remarks to the Author):

Reviewer #4 (Remarks to the Author):

Rho, an ATP-dependent helicase, binds to ssRNA and/or RNA polymerase (RNAP), translocates along the RNA, and applies mechanical force to RNAP to trigger transcription termination. Under permissive growth conditions, tight coupling of transcription and translation prevents Rho from accessing its substrate RNA/RNAP due to steric hindrance by the ribosome. The manuscript explores how Rho maintains specificity under cellular stress, where inefficient translation might allow lethal excessive Rho termination. The manuscript posits that oligomerization is crucial, a conclusion initially unconvincing due to incomplete data in the original manuscript. Unfortunately, the conclusion is still not sufficiently supported by the data. Although the G150D protein forms filaments, there is no evidence that the WT protein forms the same filaments. In the last review, we suggested making mutations that disrupt the filament interface to test if they alter the pelleting of WT Rho. We also add here that biological confirmation would be that these mutations might rescue the G150D phenotype in the presence of MUP (Fig. 4). In the absence of structure-guided function experiments, we are not convinced that the filaments observed in the G150D structures occur with WT Rho, and we cannot recommend acceptance in Nat. Comm. without mutational studies. Below is a more in-depth review.

We could not find a request for mutations disrupting the filament interface in the original review and thus did not test such mutations during the first round of revisions. To address the reviewer's concerns, we introduced E106A+Q378A double substitutions at the filament interface designed to remove stabilizing contacts. We found that the double mutant was significantly defective in filamentation, as evidenced by pelleting assays in the presence of ADP (40% as compared to 67% for the WT Rho). This result has been added as Supplementary Fig. 8. Although filamentation was not abolished, we point out that Rho filaments are likely stabilized by many weak interactions that cooperatively take effect as they are repeated at the multiple interfaces along the filament, making them relatively impervious to disruption. Collectively, our structure-guided mutational analysis demonstrates that residue substitutions at the interface observed in the G150D and G152D filament structures can either stabilize (two sensor Cys residue pairs) or destabilize (E106A+Q378A) filaments when introduced into the WT Rho.

In the revised manuscript, we stated that we were able to visualize filaments on EM grids and obtain their structures only when the connector substitutions and the C-terminal His tag were present (lane 258):

Formation of stable (for cryoEM analyses) N-to-N dimers required prior crosslinking, and long ρ filaments could be directly observed on EM grids only if additionally stabilized by the C-terminal H₈ extension and linker substitutions. Double Cys substitutions at the protomer interface stabilized filaments formed by the untagged WT ρ , enabling their detection upon crosslinking; in contrast, substitutions that removed inter-protomer contacts destabilized ρ filaments (Supplementary Fig. 8).

We are also puzzled by the reviewer's statement that "there is no evidence that the WT protein forms the same filaments" as G150D. Although we do not have a cryoEM structure of the WT filaments, we show that (in the presence of ADP) the crosslinking pattern of WT and G150D variants is identical (revised Fig. 2c) and that both proteins aggregate in the pelleting assay (Fig. 2a), a standard test in the filaments field. We also used structure-guided crosslinking to show that both the His8 tag and G150D substitution favor the filament formation. The reviewer does not question the data obtained with the His-tagged G150D variant but the untagged G150D protein does not form filaments on the grids, and the untagged WT Rho certainly would not be expected to form filaments under the same conditions. Although we acknowledge the power of visual images obtained by microscopy, we think that our biochemical data strongly support a notion that untagged Rho can form filaments stabilized by nucleotides and mutations in the connector region.

*The revised text states (lane 122): Critically, patterns of BMOE-crosslinked products were nearly identical for the WT and G150D ρ^{X1} variants in the presence of ADP (Fig. 2c). Collectively, our results demonstrate that WT *E. coli* ρ can form filaments in the presence of ADP (Fig. 2a, c, d). While ρ proteins with residue substitutions in the connector region form filaments even in the absence of ADP, the fraction of filaments increases three-fold, from*

18 to 56 %, upon the addition of ADP (Fig. 2d).

The new manuscript's main finding reveals that under specific conditions (+ADP, with a C-terminal His tag), mutant Rho proteins can form extensive filaments visible via electron microscopy. Single-particle cryoEM analysis shows these filaments locking G150D Rho in conformation, likely rendering it inactive. In addition, in the revised manuscript, Rho mutants crosslinked with pppGpp reveals a novel N-to-N structure that is distinct from the C-to-N filament structure observed, consistent with the distinct biochemical properties of Rho incubated with pppGpp versus ADP.

While no extended filaments were seen under other conditions, further biochemical and genetic work supports that wild-type (WT) Rho can reversibly oligomerize under specific conditions. Sensor cysteines capable of crosslinking only in the filament conformation help validate the cryo-EM structure's relevance. Additionally, the consistent nucleotide-dependence of Rho's pelleting and oligomerization supports dynamic oligomerization as a functional switch during cellular stress.

However, the His tag data seem extraneous to the biological findings and could confuse general readers of Nature Communications. We noted this in our earlier review but the authors argue that this artifact is important because others in the field use His-tag protein. We thus request that any biochemical data showing His-tagged variants be moved to the supplement, while the main figures should only show data from non-His-tagged protein. We believe this will allow readers to focus on the biological relevance of the findings in the main text, while permitting specialists in the field to dig deeper into His-induced changes in the supplement. Currently, WT and His-tagged data are intermingled in the main figures such that it becomes confusing to understand what the biologically-relevant versus potentially artifactual protein concentrations are.

The figures also require improved labeling and legends. Notably, each panel displaying a model should clearly indicate the protein and observed ligand, including source references.

We are grateful for the detailed and thoughtful comments by the reviewer and did our best to incorporate their suggestions into the revised manuscript. We constructed and characterized mutants at the filament interface, rearranged/added figures, added details to figures and their legends, provided raw data, and made changes to the manuscript text to clarify our data interpretation.

Recommendations for Revision:

1. Label figures 1-4 and clarify legends to enhance comprehensibility.

Revised to clarify and add detail

2. Omit or move His tag data to supplement to streamline the narrative. Main figures should speak directly to biological relevance, rather than technical details of interest to specialists.

In the revised manuscript, we moved all biochemical data collected with the His-tagged proteins (except for their cryo-EM structures) to the supplement. Pelleting assays from Fig. 2a were moved to the Supplementary Fig. 6b and crosslinking data from Figure 2ef - the Supplementary Fig. 6a,f.

3. Detail the genetic and biochemical methodologies supporting reversible oligomerization and its physiological relevance.

We added more details to the main text, figure legends, and the Methods section describing assays and justification for testing the oligomers disassembly.

Specific Figure Comments:

1. Figure 1:

o In the accompanying text and Figure Title, please incorporate sooner that a His8 tag is required to observe the filaments.

We incorporated the information about the His tag when introducing the filaments structures in the text and in the figure legend.

o (b) Please include the same WT-Rho experiment for comparison to demonstrate that G150D/G152D are also required for filamentation, not just the His tag.

We assume that the reviewer meant an image of negatively stained WT His-tagged Rho. We did not collect data with the tagged WT Rho at the time of the G150D/G152D mutants' analysis. We used tagged mutants from a previous study because the His8 tag does not affect Rho properties in vitro in assays performed at low Rho concentrations. Once we realized that the tag is located at the domain interface, we sought to ensure that the tag was not sufficient for Rho oligomerization. We thus focused on biochemical assays to address the effects of His tag on Rho aggregation. Our results demonstrate that His tag promotes filament formation at low Rho concentrations, whereas G150D substitution promotes filament formation in the absence of ADP at physiological concentrations of Rho and ADP. In the original manuscript, all key experiments were done with untagged proteins. We used His-tagged proteins for comparison in vitro and for ex vivo BMOE crosslinking when expressing Rho from plasmids because we could detect the proteins with a commercial fluorescent stain. We do not have a lot of Rho antibodies and they are not commercially available, it was a gift.

The His tag was instrumental in capturing the filaments but is undeniably artifactual. We therefore do not think that images of WT His-tagged Rho on grids would provide valuable information. To address the concerns of the reviewer and to incorporate the Editor's suggestions, we indicated the presence of the tag in every experiment in the main text and in the Supplementary data and added several statements to the Results and Discussion sections to emphasize the contribution of the octahistidine tag to Rho oligomerization:

Lane 115 Our failure to detect filaments on grids with untagged proteins suggested that the tag, which is adjacent to $\alpha 16$ making key filament-stabilizing contacts (see above), may favor filament formation.

Lane 117 ...BMOE-mediated crosslinking showed that that G150D^{X1}•H₈, but not G150D^{X1}, formed filaments at 50 nM (Supplementary Fig. 6f); and pelleting assays detected filament formation at lower concentrations in the presence of the H₈ tag (compare Fig. 2a and Supplementary Fig. 6b).

Lane 258 ...long p filaments could be directly observed on EM grids only if additionally stabilized by the C-terminal H₈ extension and linker substitutions.

o (c) Unclear contents; please specify map versus model in the figure and legend. In addition, we recommend that a locally filtered map be provided in Fig. 1C to better support the sections of the model built at the periphery.

Fig. 1c shows CryoEM reconstruction (left) and combined cartoon/semitransparent surface model (right) of G150D•H₈ filaments. In both the reconstruction and the model, stacked p hexamers are colored in different shades of cyan. *The density shown in Fig. 1c is a locally filtered map. We modified the legend to state:*

c, CryoEM reconstruction of G150D•H₈ filaments. See Supplementary Figs. 1 and 2 for structural analysis of p G152D•H₈ filaments. Left, 3D reconstruction, a locally filtered map; Right, model of the G150D filament in cartoon and semitransparent surface representation.

o (e), (f), (g) Specify which structures are depicted, and show corresponding maps for H-bond claims. Some of these structures are not from this paper, right? If so, are they crystal, cryoEM, same buffers and concentrations of ligands?

The corresponding 3D reconstructions were included to Fig. 1 e-g. We now clearly specify the structures in the figure legend. Only the structure of WT ρ bound to ATP (PDB ID: 6WA8) is not from this study. This structure was solved by X-ray crystallography at a resolution of 3.3Å, which is similar to the resolution of the structures in this study. The structure corresponds to the endogenous *E. coli* ρ protein and was isolated as a contamination during the purification of a recombinant ABC binding cassette transporter overexpressed in *E. coli*.

2. Figure 2:

o (a) Remove distracting His-tag data after noting it in the begin and consider a supplementary figure or appendix for it. Is there un-tagged G150D data that could be provided for comparison with WT instead?

We moved Fig. 2ef into the supplement (Supplementary Fig. 6) and used a pelleting assay with the untagged G150D to Fig. 2a instead.

o (b) Can't clearly see the text that is overlaid onto the figure

Modified.

o (d) This table is difficult to digest; bar graph visualizations would be more helpful. In addition, are these with tagged or untagged proteins? No His tag is indicated, yet the ADP concentration used was 2 mM (while in the legend for 2A, it is said that you used 2.5 mM ADP for untagged Rho).

Untagged proteins were used in Fig. 2d; throughout the manuscript, all tagged proteins are indicated with H₈. In the revised manuscript, we presented these data as bar graphs and also included a trace analysis (Fig. 2c) to demonstrate that WT and G150D proteins form nearly identical crosslinked products in the presence of ADP.

We used 2.5 mM ADP for pelleting assays of untagged Rho and 2 mM ADP for crosslinking. We have repeated the pelleting assay (now shown in Fig. 2a) with 2 mM ADP.

o (e), (f) Move to supplement for specialists who want clarity on the His tag effects

Moved to Supplementary Fig. 6. His tag can be responsible for oligomerization of many other proteins, as was first (to our knowledge) shown in 1996 by Wild and Filutowicz (PMID: 10698267) – this is why we initially kept the His-tag data in the main text: many people use His-tagged proteins, not just Rho specialists.

3. Figure 3:

o Can experimental density be provided for the previously published WT-ATP structure in to support their claims in Fig. 3A?

3D reconstructions are now shown for each structure.

o Why the different colors in a? The use of the same colors between 3a and 3c makes it seem as if the schemes are related, but they are not. It would be more clear to provide text labels at the top of the figure with the specific Rho allele and ligand written next to one another.

We now provide text labels at the top of each panel in Fig. 3a to specify which Rho and ligand is shown. We also changed the color in Fig. 3a, panel 1 to avoid confusion.

o This figure shows that ADP pre-incubation inhibits Rho activity. These are nice data. However, raw gel images were not provided for Fig. 3b and Fig. 3c in the accompanying Excel file (we do appreciate seeing the replicates for the other crosslinking and sucrose gradient experiments). Please provide these.

We included a gel image in the figure and also provided raw TLC plate and gel images for Fig. 3b and Fig. 3c, respectively, in the Source Data.

o Please specify how the p-values were calculated in the Figure Legend.

Added: Two-tailed T-test assuming unequal variance was used to calculate p-values.

4. Figure 4:

o Clarify the experiment and what the y-axis represents in Fig. 4 c, d and e. It is not clear that these are in vivo pelleting assays, rather than sucrose gradient fractionation experiments as in Fig. 4a.

We modified the Fig. 4 legend to include details and justification.

5. Figure 5:

o Where did the structure in Fig. 5c come from? Please provide PDB ID if previously published.

We now provide a more detailed figure legend for Fig. 5c. The structure was determined in this study, which is now stated in the legend.

6. Figure 6:

o PDB ID missing in Figure Legend

PDB ID (6WA8) is now provided in the legend.

Text:

- Lines 122-125: "In contrast, in the ATP-bound state, the R353-W381 contact is more dynamic, as we observed distinct orientations of these two residues within the hexamer. The cation- π interaction must be disrupted during ring closure, and higher flexibility of R353 and W381 likely facilitates the underlying conformational changes."

The argument that R353 and W381 are more dynamic/flexible when ATP is bound is currently not supported. The rightmost panel of Fig. 3A implies that R353 and W381 sample one conformation in the ATP-bound state (albeit a different conformation than the ADP-bound states), while the text implies that multiple conformers can exist when ATP is bound. Please clarify this point.

The structure of WT Rho bound to ATP (6WA8) was previously determined by X-ray crystallography at a resolution of 3.3 Å. While density for W381 is well-defined at each of the protomer interfaces, the density for R353 is either not resolved, fuzzy or discontinuous. Therefore, the conformation of the R353 side chain cannot reliably be built. Weak or fuzzy density indicate higher flexibility or multiple conformations of R353. In contrast, reconstructions of the G150D filament and WT ρ bound to ADP show clear experimental densities for both, the W381 and R353 side chains at each protomer interface, respectively, indicating that the conformation is restrained by the strong cation- π interaction between R353 and W381. Therefore, we suggest that in the ATP-bound state residues R353 and W381 are more dynamic and that not every R353:W381 pair of the hexamer is simultaneously involved in forming a strong interaction. We now revised the main text to clarify our arguments:

Lane 131 In contrast, in the ATP-bound state, the R353-W381 contact **seems to be** more dynamic. **While density for W381 is well-defined at each of the protomer interfaces, density for R353 is either not resolved or discontinuous, indicating higher flexibility or multiple conformations of R353.**

- Please revise the title of PDB entries to clearly indicate whether Rho is His-tagged; whether Rho is mutated (for instance, in the BMOE structure, presumably the mutant cysteine is present?) and specify bound ligands (e.g. does the G152D structure also contain ADP?)

The entry title of the BMOE structure has already been changed to “Structure of the BMOE-crosslinked transcription termination factor Rho in the presence of ppGpp; S84C/M405C double mutant”.

We changed the entry titles of the G150D and G152D mutants from “Bacterial transcription termination factor Rho G150D/G152D mutant” to “Bacterial transcription termination factor Rho G150D mutant; C-terminal 8xHis-tag” and “Bacterial transcription termination factor Rho G152D mutant; C-terminal 8xHis-tag”.

Reviewer #5 (Remarks to the Author):

Wang et al. have substantially improved their manuscript upon incorporating new data, reorganizing the manuscript structure, and tuning down the most hypothetical claims. What is clear from the work is that *E. coli*'s Rho is able to form higher-order, inactive oligomers upon the introduction of a G-to-D substitution in the connector loop between Rho's NTD and CTD or upon binding to ADP or ppGpp. The authors provide cryoEM structures of both Rho filaments and Rho N-to-N double-ring dodecamers that illustrate the versatility of Rho hyper-oligomerization in vitro (or ex vivo under conditions of Rho overexpression from a plasmid). The hyper-oligomers appear to be rather unstable and require either a His8-tag or chemical cross-linking to be observed by cryoEM. Under conditions of translational or transcriptional stress, the authors find Rho in ultracentrifugation pellets from cell lysates and tentatively connect these aggregates to the hyper-oligomers observed in vitro. Despite their efforts, the formation of the hyper-oligomers under physiological conditions remains uncertain and further (likely difficult) work will be required to rigorously establish this point and rule out alternative scenarios. Nevertheless, I think that the authors provide an extensive and intriguing body of work that challenges our current conception of Rho-dependent regulation, and that will likely prompt follow-up studies from groups interested by this topic.

We thank the reviewer for their positive assessment of our efforts to improve the manuscript during revisions, as well as for pointing out challenges in studying the physiological role of Rho oligomers in the cell. We are actively working to investigate Rho regulation during stress but studies of slowly (or not at all) growing bacteria are more technically difficult and time consuming than studies of "happy" bacteria. We made textual changes to the manuscript and incorporated new data to address the reviewer's comments.

Specific points:

1) Define 'rut' first time used (lane 31).

Defined.

2) Remove mention of the RNAP-centered scenario as the favored scenario (lanes 35-36). This point is hotly debated and, in any case, favoring this or other scenarios has absolutely no consequence for the present work.

Removed.

3) The description of the structural details on pages 4-5 is difficult to follow. I suggest that the authors provide positions of the various loops/helices they discuss, either on the diagram of fig1a (where only helix alpha 6 is depicted) or in fig S3a.

All discussed elements have been added to Supplementary Fig. 3a.

4) More generally, I think the clarity of the manuscript would benefit from mentioning more frequently the figure panels where the readers can find the reported findings.

We went through the manuscript and added references to the relevant figure panels.

5) The in vitro transcription termination data now reported in figure 3c provide an important control BUT: i) it is customary to provide a representative gel image that will help experts assess the nature of the effects. For instance here, how the inhibitors affect the distribution of signals between the various termination sites, whether the overall transcription yield is also affected, etc. ii) why has ppGpp not been tested alongside the other nucleotides?

We added a gel panel to Fig. 3c and provided TLC plate and gel images for Fig. 3b and Fig. 3c in the Source Data.

ppGpp was tested at the same time but not added to Fig. 3 because we have not yet introduced ppGpp at this point. The effect of ppGpp is complex because, unlike ADP and GDP, ppGpp directly binds to RNAP and stimulates pausing, thereby increasing both intrinsic and Rho-dependent termination. E.g., the average nucleotide addition rate by E. coli RNAP is decreased by 30% at 30 μ M ppGpp and Rho termination is increased at 25 μ M ppGpp (PMID: 22210857), whereas here we used 1 mM ppGpp. Our results show that ppGpp effect on ATPase activity is small (Supplementary Fig. 11b) and its effect on RNAP apparently dominates in the in vitro transcription assays, which have been included in the revised manuscript as Supplementary Fig. 11c. The description of these results reads:

The effects of ppGpp on termination are complex because, unlike ADP and GDP, ppGpp directly binds to RNAP³³ and stimulates pausing, thereby increasing both intrinsic and ρ -dependent termination. The average nucleotide addition rate by E. coli RNAP is decreased by 30 % at 30 μ M ppGpp and ρ termination is increased at 25 μ M ppGpp³⁴ whereas here we used 1 mM ppGpp. Our results show that ppGpp effect on RNAP apparently dominates in the in vitro transcription assays, reducing readthrough at the λ tR1 signal with or without ρ (Supplementary Fig. 11c).

6) The statistical test(s) used in Fig 3b,c is(are) not mentioned.

A statement “Two-tailed T-test assuming unequal variance was used to calculate p-values” has been added to the figure 3 legend

7) Lane 167: “...since (ρ)ppGpp binding alone is not sufficient to promote pelleting...”. Where is the supporting data in the current manuscript?

We showed that ppGpp does not induce a shift of His-tagged ρ to the pellet fraction in Supplementary Fig. 14b. We also included a pelleting assay with the untagged protein as Supplementary Fig. 14d in the revised manuscript.

8) Lanes 208-214: The authors use sucrose gradient as an independent means to detect formation of oligomers with native Rho (instead of the Cysteine-modified variants used in BMOE cross-linking assays). There is a strong pellet signal, suggesting oligomerization in presence of ADP (figS12b), but the protein used is Rho-His8, not native Rho. Is the same signal observed with native (untagged) Rho? Again, this questions the relevance of the authors' findings as potential lab curiosities.

The same signal has been observed for untagged Rho. The sucrose gradient result of untagged Rho + ADP is now provided in Supplementary Fig. 14c.

Response to Reviewer Comments

Reviewer comments are repeated in **bold**, our responses are in *italics*. Passages added to/modified in the revised manuscript in response to reviewers' comments are indicated in **red**.

Reviewer #1 (Remarks to the Author)

The points raised in my previous report have been satisfactorily addressed.

No response is needed.

Reviewer #5 (Remarks to the Author)

Review NCOMMS-23-60851B

Wang et al have satisfactorily addressed my concerns in their revised manuscript. The manuscript would further benefit from the following minor changes:

1) More consistently indicating which version (H8 or untagged) of Rho was used in each of the figure panels. This is not always obvious without digging deep in the paper and even sometimes remains uncertain.

As suggested, we have updated the figure legends of Figures 1 – 6, Supplementary Figures 1, 2, 4, 5, 6, 9, 11, 12, 13.

2) Indicating the degree of conservation between Rho factors from *E. coli* and *P. aeruginosa* (~81% identity and ~89% similarity if I am not mistaken), e.g. in lane 93. Figure 7a could also be modified to indicate that the top schematic is for both *E. coli* and *P. aeruginosa* Rho factors.

*We provided this information in Supplementary Figure 7. We added the pairwise alignment of the filament interfaces in *E. coli* and *P. aeruginosa* Rhos as panel a. We also stated that the two proteins are 81.6% identical and 92.6% similar, as calculated by EMBOSS Needle (PMID: 38597606), which uses Needleman-Wunsch algorithm and BLOSUM62 substitution matrix.*

We also modified the top schematic of Figure 7a as suggested by the reviewer.

I congratulate the authors for their extensive, somewhat provocative work and for their efforts at improving their manuscript during the revision stages. I am eager to see if future work will indeed prove unambiguously the physiological relevance of their “Rho polymerization under stress” model.

Reviewer #6 (Remarks to the Author)

Many hexameric ATPases (in particular AAA+ ATPases) are known to have a tendency to form higher order oligomers and filaments although it has been shown that the active form is a non-planar hexamer. It has been proposed in the past that the inactive higher (than hexamer) order oligomers could be a storage form although this was not observed in cells. Given the lack of functional activities *in vitro* and inability to observe in cells, the idea of storage form could not be adequately tested. Furthermore, most of the higher order oligomers were observed *in vitro* with mutants or with high concentration of one particular type of nucleotide (such as ADP or AMPPNP), making it more difficult to argue this is physiologically relevant in cells, where there will be a mix of ATP and ADP.

The manuscript here reports *in vitro* and *in vivo* data on Rho hexamer helicase, arguing that the filaments and dodecamers they observe *in vitro* (with mutants or ADP or pppGpp) could represent inactive forms in cells. However, these fall into similar conundrum as data on higher order oligomers shown previously. It is plausible but difficult to prove. The manuscript presents various *in vitro* and *in vivo* data. But the data are sometimes inconsistent, partly due to the varying effects of his-tags, mutations, nucleotides and *in vitro* vs. *in vivo*.

We agree that we did not unambiguously demonstrate the physiological relevance of Rho hyper-oligomerization, which is a daunting task for many proteins, particularly those that are essential. We added a discussion of these challenges on lines 278 – 285, as follows

Although diverse proteins implicated in cellular metabolism have been shown to form filaments, the scarcity of methods that enable direct detection of native proteins within the complex intracellular environment poses a significant challenge to establishing the physiological relevance of filamentation³⁸. Our findings that an eight-residue tag dramatically alters ρ oligomerization properties (Supplementary Fig. 6f) illustrate why a common approach, fusing a large fluorescent protein to the target to visualize aggregation in cells³⁸, may lead to artifacts. Future studies are necessary to determine the composition and structure of cellular ρ aggregates, reveal whether they represent transient storage depots or dead-end complexes targeted for disposal, and identify cellular factors that modulate ρ oligomerization.

However, we think that our data are internally consistent. We show that substitutions in the linker and the His tag at the C-terminus of Rho stabilize the ADP-bound filaments. Nonetheless, the wild-type Rho forms similar filaments, as observed by crosslinking. We note that we tested Rho oligomerization at physiological concentrations of Rho throughout the manuscript and demonstrated hyper-oligomerization in a mixture of nucleotides [including ATP, ADP, ppGpp] present at physiological concentrations, as shown in Supplementary Fig. S11d.

The authors make little attempt to provide/reconcile mechanistic/molecular basis for the data. For example, there are multiple paragraphs/subsections on the mutant G150D, and the requirement of ADP/ATP on filaments/oligomer formations. ADP is required for G150D filaments formation in cells as in their pelleting assays, but not required *in vitro*.

*We apologize for the confusion. Although G150D Rho forms filaments in the absence of nucleotide, ADP significantly increases the fraction of G150D filaments *in vitro* (Figure 2 c,d and Supplementary Figure 11a). We didn't propose that ADP is required for G150D filamentation *in vivo*, as we have no direct data to support this model. Our conclusion that the increased relative [ADP] promotes filamentation is based on the *in vitro* analysis at defined nucleotide concentrations and *in vivo* analysis under "stress" conditions [stationary phase, antibiotic treatment, Mg stress] when ADP/ATP ratios and ppGpp levels are known to rise. Focusing on the wild-type Rho, we find that, under these conditions, Rho forms large oligomers and that the G150D substitution promotes oligomerization, consistent with our *in vitro* observations. The G150D rho strain is also hypersensitive to an*

increase in ppGpp levels induced by the addition of mupirocin or expression of active RelA. Based on these results, we speculated that stress-induced nucleotides (ADP, ppGpp) promote Rho hyper-oligomerization.

In the structures, they show the importance of bound nucleotide in stabilising the filaments, so one would expect ADP does promote filaments formation in vitro. The authors should try to resolve the discrepancy on whether nucleotides (either ATP or ADP) promote filament formation.

There is no discrepancy. We have shown that ADP promotes filamentation of both wild-type and G150D variants (Figures 2, 5, Supplementary Figure 11). In contrast, ATP can reverse the filamentation/oligomerization (Figure 5d). Our structures provide evidence for the molecular basis of higher oligomer formation. Although we could not detect filaments on the grids for wild-type Rho, our structures clearly revealed nucleotide-dependent conformational changes in the dynamic interaction network of the Rho hexamer and more pronounced at the nucleotide-binding pocket. We suggest that these changes lead to a more stably bound ADP and tightening of the interaction network at and close to the nucleotide-binding pocket, which otherwise appears very flexible. This could stabilize Rho in an open conformation, a prerequisite for the assembly of higher oligomeric states. In this regard, ADP-dependent stabilization of the open state would facilitate filament formation, in complete agreement with the pelleting and BMOE-crosslinking assays.

Could it be due to higher protein concentrations in in vitro experiments? This could be tested the in vitro effects with varying protein-concentrations. Furthermore, they seem to argue ADP stabilises open ring conformation, rather than filaments, causing further confusions.

Our in vitro experiments were done at physiological concentrations of Rho.

As noted above, ADP stabilizes open ring conformation and changes Rho promoter interface, promoting filament formation.

Overall, it is important to demonstrate the filaments can form for WT proteins. Even if the filaments are unstable, the authors should try crosslinking (with their cysteine mutants) before putting onto EM grids (either negative stain or cryo). Direct visualisation is important even though the numbers of filaments might be low and from which it will be difficult to obtain high resolution structures. Ideally its relevance in vivo should be demonstrated. Based on the structures, are there potential mutations that could interfere with higher-order oligomer formation without interfering with active open ring formation? If so, introducing these mutations in vivo might demonstrate the importance of these storage/inactive forms in cellular activities.

Although we understand the appeal of “direct visualization”, we have shown that the wild-type and G150D Rho variants have nearly identical patterns of dodecamers and higher-order oligomers captured by BMOE crosslinking (Figure 5).

We demonstrated that a double mutation that would eliminate a hydrogen bond/salt bridge stabilizing two Rho rings within the filament structure (E106A+Q378A) compromises filamentation (Supplementary Figure 8). Again, we cannot unambiguously prove the physiological relevance of Rho filamentation, and we made this limitation clear in lines 278-285; see response to the first comment.

The data on wildtype Rho aggregation in cells under stress do not necessarily support that they form filaments or higher order oligomers, as there are so many processes that have been affected in these cells (translation, transcription and others affected by antibiotics). Since Rho interacts with many other proteins and RNA, which can also aggregate under stress, the observed Rho aggregation could therefore be indirect.

We agree. Our BMOE crosslinking assays demonstrate Rho filamentation *in vitro*, but we do not know the structure or the composition of Rho aggregates in the cell. We stated on line 229 that “**p aggregation in stressed cells (Fig. 4c) is promoted by yet-unknown cellular factors**” and pointed out that Rho pelleting is most likely not due to (p)ppGpp, at least not alone.

We revised the title and the abstract to downplay the “physiological relevance” of our findings.

Minor comments:

Why did ppGpp structure show open ring conformation even though it showed similar levels of dodecamer formation as with pppGpp? It is unclear what the effects of ppGpp or pppGpp are.

We obtained the Rho-pppGpp dodecamer structure with the Rho^{X1} Cys-mutant after BMOE crosslinking. We do not think that ppGpp and pppGpp have different effects on Rho but rather that the dodecamer is not stable under conditions of grid preparation. Our crosslinking results show that pppGpp and ppGpp promote oligomerization. According to our structure, both should stabilize open ring conformation. (p)ppGpp can also remodel Rho protomer interface (Figure 6).

Since G150D induces major conformational changes in the loop, would this be possible in WT proteins? If this is possible, how would stress induce the changes that stabilise the conformation?

Yes, we think that wild-type Rho could adopt loop conformations as observed in G150 or G152 filaments. The loops of the G150D and G152D variants are changed by the G-to-D mutations, as the introduced aspartate side chains engage in hydrogen-bond or salt-bridge interactions to residues nearby (to Y197 in the case of D150) or within the same loop (to R149 in the case of D152). The same interactions are of course not possible via glycine residues at position 150 or 152, as adequate side chain functionalities are missing. Nevertheless, simple modeling (exchanging D150 or D152 with G in the filament conformations) shows that wild-type Rho could likewise adopt loop conformations as observed in the G150 or G152 filaments without any steric hindrance. The loop conformation observed in open wild-type Rho hexamers (e.g., in our wild-type Rho hexamer structure in complex with ADP) is stabilized by hydrogen-bond interactions between the side chain of N151 and the side chain and backbone amide of S153. These interactions are broken in the loop conformations observed in the filament structures, but with G residues at positions 150 or 152 would not be compensated by alternative contacts (as observed with D150 or D152). Therefore, wild-type Rho could in principle adopt D150/D152 filament-like loop conformations, and these loop conformations may be stabilized upon filamentation by nearby conformational changes in the nucleotide-binding pocket (such as a hydrogen bond between the D156 side chain and the adenine N6, which we observe in the D150 and D152 filaments but not in the wild-type Rho-ADP structure, see Fig. 6a). On the other hand, we do not discern an obvious reason why the G150/152 loop conformations should be absolutely required for Rho to form filaments.

To clarify these points, we added a shorter version of this description, focused on the comparison to G150D filaments, to the main text, lines 250-260.

In G150D filaments, an altered $\alpha 5/\alpha 6$ loop conformation is stabilized by a hydrogen bond between the D150 side chain and Y197 (Fig. 1e). Can the wild-type ρ adopt this state? Our modeling shows that reverse engineering of G150 into the G150D^{F-ADP} structure does not alter the observed loop conformation. In ADP-bound open wild-type ρ hexamers, the loop is stabilized by hydrogen-bond interactions between the side chain of N151 and the side chain and backbone amide of S153. These interactions are broken in the filaments and would not be compensated by alternative contacts (between D150 and Y197) with the G residue at position 150. Instead, upon filamentation, the loop may be stabilized by conformational changes in the adjacent nucleotide-binding pocket (e.g., a hydrogen bond between the D156 side chain and the adenine N6, which we observe in the G150D filaments but not in the wild-type ρ -ADP structure; Fig. 6a). Importantly, however, although our modeling

suggests that the wild-type ρ can adopt a filament-like $\alpha 5/\alpha 6$ loop conformation, we do not discern an obvious reason why these changes should be required for ρ to form filaments.

We do show that ADP stabilizes the otherwise flexible interaction network at the nucleotide-binding pocket, which also affects inter-subunit contacts. We suggest that these changes could lead to an increase in the helical rise of the hexamer, ultimately allowing additional Rho monomers to join the ring and form higher oligomers. While we do not know the exact conditions under which filaments form in vivo and whether conformational changes in the loop region accompany this hypothetical transition, we discuss a recently reported structure in which wild-type Rho forms higher-order oligomers without extensive changes in the loop region.

If filaments exist in cells. How are the stable filaments converted back to active hexamers?

The filaments and oligomers are reversible in vitro (Figure 5d), and Rho-containing aggregates are reduced upon recovery from antibiotic stress (Figure 4e). We hypothesize that, once the stress is relieved, ATP can bind to Rho to promote the filaments' dispersal.

On lines 273-275, the revised text states “nucleotide-stabilized inactive ρ oligomers must be intrinsically labile to ensure their facile reversal once growth resumes and ATP/ADP ratio rises; in agreement with this idea, ρ oligomerization is partially reversible in the presence of ATP γ S *in vitro* (Fig. 5d)”.